# TOPOLOGY MATTERS IN FAIR GRAPH LEARNING: A THEORETICAL PILOT STUDY

## ABSTRACT

Recent advances in fair graph learning observe that graph neural networks (GNNs) further amplify prediction bias compared with multilayer perception (MLP), while the reason behind this is unknown. In this paper, we conduct a theoretical analysis of the bias amplification mechanism in GNNs. This is a challenging task since GNNs are difficult to be interpreted, and real-world networks are complex. To bridge the gap, we theoretically and experimentally demonstrate that aggregation operation in representative GNNs accumulates bias in node representation due to topology bias induced by graph topology. We provide a sufficient condition identifying the statistical information of graph data, so that graph aggregation enhances prediction bias in GNNs. Motivated by this data-centric finding, we propose a fair graph refinement algorithm, named *FairGR*, to rewire graph topology to reduce sensitive homophily coefficient while preserving useful graph topology. Experiments on node classification tasks demonstrate that *FairGR* can mitigate the prediction bias with comparable performance on three real-world datasets. Additionally, *FairGR* is compatible with many state-of-the-art methods, such as adding regularization, adversarial debiasing, and Fair mixup via refining graph topology. Therefore, *FairGR* is a plug-in fairness method and can be adapted to improve existing fair graph learning strategies.

## 1 INTRODUCTION

Graph neural networks (GNNs) (Kipf & Welling, 2017; Veličković et al., 2018; Wu et al., 2019) are widely adopted in various domains, such as social media mining (Hamilton et al., 2017), knowledge graph (Hamaguchi et al., 2017) and recommender system (Ying et al., 2018), due to remarkable performance in learning representations. Graph learning, a topic with growing popularity, aims to learn node representation containing both topological and attribute information in a given graph. Despite the outstanding performance in various tasks, GNNs still inherit or even amplify societal bias from input graph data (Dai & Wang, 2021). The biased node representation largely limits the application of GNNs in many high-stake tasks, such as job hunting (Mehrabi et al., 2021) and crime ratio prediction (Suresh & Guttag, 2019). Hence, bias mitigation that facilitates the research on fair GNNs is in urgent need.

In many real-world graphs, nodes with the same sensitive attribute (e.g., ages) are more likely to connect. For example, young people mainly make friends with people of similar ages (Dong et al., 2016). We call this phenomenon "topology bias". Even worse, in GNNs, the representation of each node is learned by aggregating the representations of its neighbors. Thus, nodes with the same sensitive attributes will be more similar after the aggregation. To get a sense, we visualize the topology bias for three real-world datasets (Pokec-n, Pokec-z, and NBA) in Figure 1, where different edge types are highlighted with different colors for the top-3 largest connected components in original graphs. Such topology bias leads to more similar node representation for those nodes with the same sensitive attribute, which is a major source of the graph representation bias.

Existing bias mitigation work for GNNs is empirical via adding regularization, adversarial debiasing, or contrastive learning. These works are motivated by the fact that graph neural networks trained on graphs may inherit the societal bias in data, and the topology of graphs and the message passing in GNNs could even magnify the bias compared with multilayer perception (MLP) (Dai & Wang, 2021). However, even though fair prediction in GNN can be achieved via a fair training strategy, a

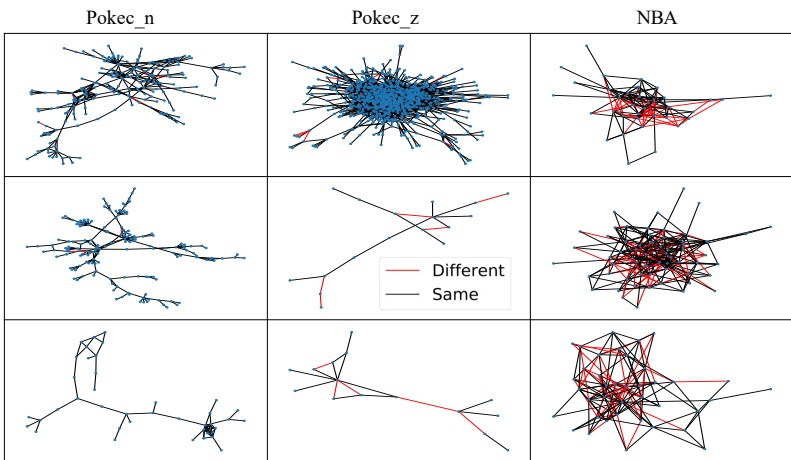

Figure 1: Visualization of topology bias in three real-world datasets, where black edges •——• and red edges •——• represent the edge with the same or different sensitive attributes for the connected node pair, respectively. We visualize the largest three connected components for each dataset. It is obvious that the sensitive homophily coefficients (the ratio of homo edges) are high in practice, i.e., $95.30\%$, $95.06\%$, and $72.37\%$ for Pokec-n, Pokec-z, and NBA dataset, respectively.

fundamental understanding of why large topology bias and message passing algorithm amplifying the bias happens is still missing. A natural question is raised:

*Can we theoretically understand why large topology bias and message passing algorithm amplify the bias from a graph data perspective?*

In this work, we move the first step to understand why large topology bias and message passing algorithm amplify the bias. Specifically, we first define the sensitive homophily coefficient to describe how likely the connected nodes are with the same sensitive attributes. Subsequently, we theoretically prove that the GCN-like aggregation in message passing inevitably accumulates representation bias for graphs with large sensitive homophily coefficients. Second, motivated by our theoretical analysis, we develop a fair graph refinement algorithm, named FairGR, to achieve fair GNN prediction via revising graph topology. More importantly, FairGR is a plug-in data refinement method and compatible with many fair training strategies, such as regularization, adversarial debiasing, and Fair mixup. In short, the contributions can be summarized as follows:

- To the best of our knowledge, it is the first paper to theoretically investigate why the GCN-like message passing scheme amplifies representation bias for large topology bias. Specifically, we provide a sufficient condition of graph data to show that GCN-like message passing amplifies representation bias.
- Motivated by our theoretical analysis, we propose a graph topology refinement method, named FairGR, to achieve fair prediction.
- We empirically show that the prediction bias of GNNs is larger than that of MLP on real-world datasets. Additionally, the effectiveness of FairGR is experimentally evaluated on three real-world datasets. The results show that compared to the state-of-the-art, our FairGR exhibits a superior trade-off between prediction performance and fairness, and is compatible with many fair training strategies, such as regularization, adversarial debiasing, and Fair mixup.

## 2 PRELIMINARIES

### 2.1 NOTATIONS

We adopt bold upper-case letters to denote matrices such as $\mathbf{X}$, bold lower-case letters such as $\mathbf{x}$ to denote vectors or random variables, and calligraphic font such as $\mathcal{X}$ to denote set. Given a matrix $\mathbf{X} \in \mathbb{R}^{n \times d}$, the $i$-th row and $j$-th column are denoted as $\mathbf{X}_i$ and $\mathbf{X}_{\cdot,j}$, and the element in $i$-th row and $j$-th column is $\mathbf{X}_{i,j}$. We use $l_1$ norm of matrix $\mathbf{X}$ as $||\mathbf{X}||_1 = \sum_{ij} |\mathbf{X}_{ij}|$. Let $\mathcal{G} = \{\mathcal{V}, \mathcal{E}\}$ be a graph with the node set $\mathcal{V} = \{v_1, \cdots, v_n\}$ and the undirected edge set $\mathcal{E} = \{e_1, \cdots, e_m\}$,

where $n, m$ represent the number of node and edge, respectively. The graph structure $\mathcal{G}$ can be represented as an adjacent matrix $\mathbf{A} \in \mathbb{R}^{n \times n}$, where $\mathbf{A}_{ij} = 1$ if existing edge between node $v_i$ and node $v_j$. $\mathcal{N}(i)$ denotes the neighbors of node $v_i$ and $\tilde{\mathcal{N}}(i) = \mathcal{N}(i) \cup \{v_i\}$ denotes the self-inclusive neighbors. Suppose that each node is associated with a $d$-dimensional feature vector and a (binary) sensitive attribute, the feature for all nodes and sensitive attribute is denoted as $\mathbf{X}_{ori} = \mathbb{R}^{n \times d}$ and $\mathbf{s} \in \{-1, 1\}^n$. $I(\mathbf{s}, \mathbf{X})$ represents the mutual information between the sensitive attribute and node features. $\mathbf{A} \odot \mathbf{B}$ represents Hadamard product for matrix element-wise multiplication.

## 2.2 Label and Sensitive Homophily Coefficient in Graphs

The behaviors of graph neural networks have been investigated in the context of label homophily for connected node pairs in graphs (Ma et al., 2021). Label homophily in graphs is typically defined to characterize the similarity of connected node labels in graphs. Here, similar node pair means that the connected nodes share the same label. From the perspective of fairness, we also define the sensitive homophily coefficient to represent the sensitive attribute similarity among connected node pairs. Informally, the coefficients for label homophily and sensitive homophily are defined as the fraction of the edges connecting the nodes of the same class label and sensitive attributes in a graph (Zhu et al., 2020; Ma et al., 2021). We also provide the formal definition as follows:

**Definition 1 (Label and Sensitive Homophily Coefficient)** *Given a graph $\mathcal{G} = \{\mathcal{V}, \mathcal{E}\}$ with node label vector* $\mathbf{y}$, *node sensitive attribute vector* $\mathbf{s}$, *the label and sensitive attribute homophily coefficients are defined as the fraction of edges that connect nodes with the same labels or sensitive attributes* $\epsilon_{label}(\mathcal{G}, \mathbf{y}) = \frac{1}{|\mathcal{E}|} \sum_{(i,j) \in \mathcal{E}} \mathbb{1}(\mathbf{y}_i = \mathbf{y}_j)$, *and* $\epsilon_{sens}(\mathcal{G}, \mathbf{s}) = \frac{1}{|\mathcal{E}|} \sum_{(i,j) \in \mathcal{E}} \mathbb{1}(\mathbf{s}_i = \mathbf{s}_j)$, *where* $|\mathcal{E}|$ *is the number of edges and* $\mathbb{1}(\cdot)$ *is the indicator function.*

Recent works (Ma et al., 2021; Chien et al., 2021; Zhu et al., 2020) aim to understand the relation between the message passing in GNNs and label homophily from the interactions between GNNs (model) and graph topology (data). For graph data with a high label homophily coefficient, existing works (Ma et al., 2021; Tang et al., 2020) have demonstrated, either provably or empirically, that the nodes with higher node degree obtain more prediction benefits in GCN, compared to the benefits that peripheral nodes obtain. As for graph data with a low label homophily coefficient, GNNs do not necessarily lead to better prediction performance compared with MLP since the node features of neighbors with different labels contaminate the node features during feature aggregation. However, although work (Dai & Wang, 2021) empirically points out that graph data with a large sensitive homophily coefficient may enhance bias in GNNs, the fundamental understanding of message passing and graph data properties, such as sensitive homophily coefficient, is still missing. We provide the theoretical analysis in Section 4.

## 3 Related Work

We briefly review the existing work on **graph neural networks** and **fairness-aware learning on graphs.** (Please refer to Appendix D for a more comprehensive discussion.). Existing GNNs can be roughly divided into spectral-based and spatial-based GNNs. Spectral-based GNNs provide graph convolution definition based on graph theory (Bruna et al., 2013; Defferrard et al., 2016; Henaff et al., 2015). Spatial-based GNNs variant are popular due to explicit neighbors' information aggregation, including Graph convolutional networks (GCN) (Kipf & Welling, 2017), graph attention network (GAT) (Veličković et al., 2018). As for fairness in graph data, many works have been developed to achieve fairness in machine learning community (Chuang & Mroueh, 2020), including fair walk (Rahman et al., 2019), adversarial debiasing (Dai & Wang, 2021), Bayesian approach (Buyl & De Bie, 2020), and contrastive learning (Agarwal et al., 2021). Much literature empirically shows that GNN or message passing may enhance prediction bias compared with MLP. However, the theoretical understanding of why such a phenomenon happens is still unclear. In this work, we take an initial step toward theoretically understanding why message passing enhances bias from a data perspective. Based on this understanding, we develop a simple yet effective fair graph refinement method to achieve better tradeoff performance. More importantly, the proposed FairGR is compatible with many fair training strategies.

## 4 TOPOLOGY ENHANCES BIAS IN GNNs AGGREGATION

For each node, GNNs aggregate their neighbors' features to learn their representation. In real-world datasets, we observe a high sensitive homophily coefficient (which is even higher than the label homophily coefficient). However, existing message-passing schemes tend to aggregate the node features from their neighbors with the same sensitive attributes. Thus, the common belief is that the message passing renders node representations with the same sensitive attribute more similar. However, this common belief is heuristic and the quantifiable relationship between the topology bias and representation bias is still missing. In this section, (1) we rectify such common belief and quantitatively reveal that **only sufficient high sensitive homophily coefficient** would lead to bias enhancement; (2) we analyze the **influence** of other graphs statistical information, such as **the number of nodes $n$, the edge density $\rho_d$, and sensitive group ratio** $c$, in term of bias enhancement.

### 4.1 SYNTHETIC GRAPH

we consider the synthetic random graph generation using contextual stochastic block model (CSBM) (Fortunato & Hric, 2016), including graph topology and node features generation with Gaussian mixture distribution. As a pilot study, we choose the most common-used graph topology with stochastic block model (SBM) and node feature with Gaussian mixture distributions in the analysis. The rationale for the choice is due to the fact that SBM is widely used to model and analyze most complex networks (e.g., social networks, World Wide Web, and biological networks) (Van Der Hofstad, 2016). [1]

**Graph Topology**    Throughout our analysis, we mainly focus on 4 characteristics of graph topology: the number of nodes $n$, the edge density $\rho_d$, sensitive homophily coefficient $\epsilon_{sens}$, and sensitive group ratio $c$. Specifically, we consider the synthetic random graph generation, including graph topology and node features generation as follows:

**Definition 2** $((n, \rho_d, \epsilon_{sens}, c)$**-graph)** *The synthetic random graph $\mathcal{G}$ sampled from $(n, \rho_d, \epsilon_{sens}, c)$-graph satisfies the following properties: 1) the graph node number is $n$; 2) the adjacency matrix $\mathbf{A}$ satisfies $\mathbf{A}_{ij} \in \{0, 1\}$ and $\mathbb{E}_{ij}[\mathbb{P}(\mathbf{A}_{ij} = 1)] = \rho_d$; 3) given connected node pair $\mathbf{A}_{ij} = 1$, the probability of connected nodes with the same sensitive attribute satisfies $\mathbb{P}(\mathbf{s}_i = \mathbf{s}_j | \mathbf{A}_{ij} = 1) = \epsilon_{sens}$; 4) the binary sensitive attribute $\mathbf{s}_i \in \{-1, 1\}$ satisfies $\mathbb{E}_i[\mathbb{P}(\mathbf{s}_i = 1)] = c$; 5) independent edge generation.*

**Node Features.**    We assume that node attributes in synthetic graph follow Gaussian Mixture Model $GMM(c, \mu_1, \boldsymbol{\Sigma}_1, \mu_2, \boldsymbol{\Sigma}_2)$. For the node with sensitive attribute $\mathbf{s_i} = -1$ ($\mathbf{s_i} = 1$), the node attributes $\mathbf{X}_i$ follow Gaussian distribution $P_1 = \mathcal{N}(\mu_1, \boldsymbol{\Sigma}_1)$ ($P_2 = \mathcal{N}(\mu_2, \boldsymbol{\Sigma}_2)$), where the node attributes with the same sensitive attribute are independent and identically distributed, and $\mu_i, \boldsymbol{\Sigma}_i$ ($i = 1, 2$) represent the mean vector and covariance matrix.

### 4.2 REPRESENTATION BIAS MEASUREMENT

To measure the node representations bias, we adopt the mutual information between sensitive attribute and node attributes $I(\mathbf{s}, \mathbf{X})$. Note that the exact mutual information $I(\mathbf{s}, \mathbf{X})$ is intractable to estimate, an upper bound on the exact mutual information is developed as a surrogate metric in the following Theorem 1:

**Theorem 1** *Suppose the synthetic graph node attribute $\mathbf{X}$ is generated based on Gaussian Mixture Model $GMM(c, \mu_1, \boldsymbol{\Sigma}_1, \mu_2, \boldsymbol{\Sigma}_2)$, i.e., the probability density function of node attributes for the nodes of different sensitive attribute $\mathbf{s} = \{-1, 1\}$ follows $f_{\mathbf{X}}(\mathbf{X}_i = \mathbf{x} | \mathbf{s}_i = -1) \sim \mathcal{N}(\mu_1, \boldsymbol{\Sigma}_1) \stackrel{\triangle}{=} P_1$ and $f_{\mathbf{X}}(\mathbf{X}_i = \mathbf{x} | \mathbf{s}_i = 1) \sim \mathcal{N}(\mu_2, \boldsymbol{\Sigma}_2) \stackrel{\triangle}{=} P_2$, and the sensitive attribute ratio satisfies $\mathbb{E}_i[\mathbb{P}(\mathbf{s}_i = 1)] = c$, then the mutual information between sensitive attribute and node attributes $I(\mathbf{s}, \mathbf{X})$ satisfies*

$$I(\mathbf{s}, \mathbf{X}) \leq -(1-c) \ln \left[ (1-c) + c \exp \left( -D_{KL}(P_1 || P_2) \right) \right]$$

$$-c \ln \left[ c + (1-c) \exp \left( -D_{KL}(P_2 || P_1) \right) \right] \stackrel{\triangle}{=} Bias(\mathbf{s}, \mathbf{X}).$$

---

[1]We leave the analysis of other random graph models or feature distributions in future work.

Based on Theorem 1, we can observe that lower distribution distance $D_{KL}(P_1||P_2)$ or $D_{KL}(P_2||P_1)$ is beneficial for reducing $Bias(\mathbf{s}, \mathbf{X})$ and $I(\mathbf{s}, \mathbf{X})$ since the sensitive attribute is less distinguishable based on node representations.

## 4.3 When Aggregation enhances the bias?

We focus on the role of message passing in terms of fairness. Suppose the graph adjacency matrix $\mathbf{A}$ is sampled for $(n, \rho_d, \epsilon_{sens}, c)$-graph and we adopt the GCN-like message passing $\tilde{\mathbf{X}} = \tilde{\mathbf{A}}\mathbf{X}$, where $\tilde{\mathbf{A}}$ is normalized adjacency matrix with self-loop. We define the bias difference for such message passing as $\Delta Bias = Bias(\mathbf{s}, \tilde{\mathbf{X}}) - Bias(\mathbf{s}, \mathbf{X})$ to measure the role of graph topology. Subsequently, we derive intra-connect and inter-connect probability in Lemma 1.

**Lemma 1** *Suppose the synthetic graph is generated from $(n, \rho_d, \epsilon_{sens}, c)$-graph, then we obtain the intra-connect and inter-connect probability as follows:*

$$p_{conn} = \frac{\rho_d \epsilon_{sens}}{c^2 + (1-c)^2}, \quad q_{conn} = \frac{\rho_d(1 - \epsilon_{sens})}{2c(1-c)}.$$

Subsequently, we provide a sufficient condition to specify the case that graph topology enhances bias in Theorem 2.

**Theorem 2** *Suppose the synthetic graph node attribute $\mathbf{X}$ is generated based on Gaussian Mixture Model $GMM(c, \mu_1, \mathbf{\Sigma}, \mu_2, \mathbf{\Sigma})$, and the graph adjacency matrix $\mathbf{A}$ is generated from $(n, \rho_d, \epsilon_{sens}, c)$-graph. If adopting GCN-like message passing $\tilde{\mathbf{X}} = \tilde{\mathbf{A}}\mathbf{X}$, bias will be enhanced, i.e., $\Delta Bias > 0$ if the bias-enhance condition holds: $(\nu_1 - \nu_2)^2 \min\{\zeta_1, \zeta_2\} > 1$, where $\nu_1 - \nu_2 < 1$ represents the reduction coefficient of the distance between the mean node attributes of the two sensitive attributes groups, $\zeta_1, \zeta_2$ mean the connection degree of two sensitive groups; the mathematical formulation is given by*

$$\nu_1 = \frac{(n_1 - 1)p_{conn} + 1}{\zeta_1}, \quad \nu_2 = \frac{(n_1 - 1)q_{conn}}{\zeta_2}, \tag{1}$$

*where $\zeta_1 = n_{-1}q_{conn} + (n_1 - 1)p_{conn} + 1$, $\zeta_2 = n_{-1}p_{conn} + (n_1 - 1)q_{conn} + 1$, the node number with the same sensitive attribute $n_{-1} = n(1 - c)$, $n_1 = nc$, intra-connect probability $p_{conn} = \mathbb{E}_{ij}[\mathbb{P}(\mathbf{A}_{ij}|\mathbf{s}_i = \mathbf{s}_j)]$, and inter-connect probability $q_{conn} = \mathbb{E}_{ij}[\mathbb{P}(\mathbf{A}_{ij}|\mathbf{s}_i\mathbf{s}_j = -1)]$.*

Based on Theorem 2, we can provide more discussion on the influence of 4 graph data in-formation of graph:the number of nodes $n$, the edge density $\rho_d$, sensitive homophily coefficient $\epsilon_{sens}$, and sensitive group ratio $c$ for bias enhancement as follows:

- **Node representation bias is enhanced by message passing for sufficient large sensitive homophily $\epsilon_{sens}$.** According to Lemma 1, for large sensitive homophily coefficient $\epsilon_{sens} \to 1$, the inter-connect probability $q_{conn} \to 0$ and intra-connect probability $p_{conn}$ keeps the maximal value. In this case, based on Theorem 2, it is easy to obtain that $\nu_1 = 1$, $\nu_2 = 0$ and the distance for the mean aggregated node representation will keep the same, i.e., $\tilde{\mu}_1 - \tilde{\mu}_2 = (\nu_1 - \nu_2)(\mu_1 - \mu_2) = \mu_1 - \mu_2$. Additionally, the covariance will be diminished after aggregation since $\zeta_1$ and $\zeta_2$ are strictly larger than 1. Therefore, for sufficient large sensitive homophily coefficient $\epsilon_{sens}$, the bias-enhance condition $(\nu_1 - \nu_2)^2 \min\{\zeta_1, \zeta_2\} > 1$ holds.
- **The bias enhancement implicitly depends on node representation geometric differentiation, including the distance between the mean node representation within the same sensitive attribute and the scale covariance matrix.** Theorem 1 implies that low mean representation distance and concentrated representation (low covariance matrix) lead to fair representation. However, GCN-like message passing renders the mean node representation distance reduction $\nu_1 - \nu_2$ and concentrated for each sensitive attribute group, which is an "adversarial" effect for fairness and the mean distance and covariance reduction is controlled by sensitive homophily coefficient.
- **The bias is enlarged as node number $n$ being increased.** For large node number $n$, the mean distance almost keeps constant since $\nu_1 \approx \frac{cp_{conn}}{(1-c)q_{conn} + cp_{conn}}$, $\nu_2 \approx \frac{cq_{conn}}{(1-c)p_{conn} + cq_{conn}}$, and $\zeta_1, \zeta_2$ are almost proportional to node number $n$. Therefore, the bias-enhancement condition can be more easily satisfied and $\Delta Bias$ would be higher for large graph data. The intuition is that, given graph

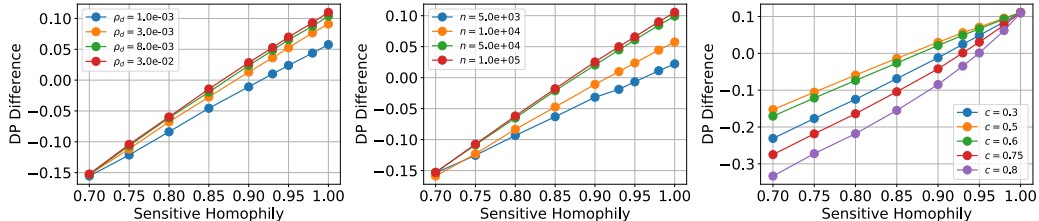

Figure 2: The difference of demographic parity for message passing. **Left:** DP difference for different graph connection density $\rho_d$ with sensitive attribute ratio $c = 0.5$ and number of nodes $n = 10^4$; **Middle:** DP difference for different number of nodes $n$ with sensitive attribute ratio $c = 0.5$ and graph connection density $\rho_d = 10^{-3}$; **Right:** DP difference for different sensitive attribute ratio $c$ with graph connection density $\rho_d = 10^{-3}$ and number of nodes $n = 10^4$;

density ratio, large graph data represents a higher average degree node. Hence, each aggregated node representation adopts more neighbor's information with the same sensitive attribute, and thus leads to a lower covariance matrix value and higher representation bias.

- **The bias is enlarged as graph connection density $\rho$ being increased.** Based on Lemma 1, inter-connection probability and inter-connection probability are both proportional to graph connection density $\rho_d$. Therefore, $\nu_1$ and $\nu_2$ almost keep constant and the distance of mean node representation is constant as well. As for the covariance matrix, message passing leads to more concentrated node representation since $\zeta_1$ and $\zeta_2$ are larger for higher graph connection density $\rho_d$. The rationale is similar to the graph node number: given node number, higher graph connection density $\rho_d$ means higher average node degree and each aggregated node representation adopts more neighbor's information with the same sensitive attribute.

- **When the sensitive attribute is more balanced (i.e.,) the bias will be enlarged.** Based on Lemma 1, it is easy to obtain that, given graph connection density $\rho_d$ and graph node number $n$, the intra-connection probability $p_{conn}$ would be high, while being low for inter-connection probability $q_{conn}$, if the balanced sensitive attribute. In other words, intra-connection probability $p_{conn}$ (inter-connection probability $q_{conn}$) monotonically decreases (increases) with respect to $|c - \frac{1}{2}|$.

## 5 LEARNING FAIR GRAPH TOPOLOGY

The above section provides a data-centric perspective (graph topology) to understanding why message passing may enhance representation bias. In other words, graph topology refinement may assist GNNs model in achieving fair and accurate prediction. However, searching for the optimal graph topology for GNNs is a non-trivial problem due to large-scale discrete optimization. In this section, motivated by the theoretical analysis, we propose **Fair Graph Refinement** method, named **FairGR**, to achieve fair prediction via learning the optimal graph topology. Specifically, we formulate the objective functions into three parts, including low sensitive homophily coefficient, high label homophily coefficient, and small topology perturbation. In this way, FairGR can explicitly reduce the topology bias while preserving useful topology information for prediction.

**Problem Formulation.** Considering binary sensitive attribute $\mathbf{s}_i \in \{-1, 1\}$ and binary classification task with label $\mathbf{y}_i \in \{-1, 1\}$ for $i$-th node, we aim to modify the graph topology to achieve low sensitive homophily coefficient $\epsilon_{sens}$, high label homophily coefficient $\epsilon_{label}$, and small topology perturbation. For sensitive homophily coefficient with the binary sensitive attribute, the determination of any two nodes with the same sensitive attribute can be obtained via $H(\mathbf{ss}^T)$, where $H(\cdot)$ is Heaviside (unit) step function, sensitive attribute vector $\mathbf{s} \in \{-1, 1\}^{n \times 1}$. Based on Definition 1, It is easy to rewrite sensitive homophily coefficient as $\epsilon_{sens} = \frac{||H(\mathbf{ss}^T) \odot \mathbf{A}||}{||\mathbf{A}||_1}$, where $\odot$ represents Hadamard product, and $|| \cdot ||_1$ denotes the entry-wise $l_1$ norms (i.e., the summation over all absolute value of elements). Similarly, label homophily coefficient as $\epsilon_{label} = \frac{||H(\mathbf{yy}^T) \odot \mathbf{A}||}{||\mathbf{A}||_1}$. As for graph topology perturbation, we can use the entry-wise $l_1$ norms of the difference as the measurement. In a nutshell, the objective function to reconstruct graph connections given sensitive attribute $\mathbf{s}$, label $\mathbf{y}$,

and graph topology $\mathbf{A}$ can be formulated as:

$$\mathcal{L}(\hat{\mathbf{A}}|\mathbf{s}, \mathbf{y}, \mathbf{A}) = \frac{||H(\mathbf{ss}^T) \odot \hat{\mathbf{A}}||}{||\hat{\mathbf{A}}||_1} - \alpha \frac{||H(\mathbf{yy}^T) \odot \hat{\mathbf{A}}||}{||\hat{\mathbf{A}}||_1} + \beta||\hat{\mathbf{A}} - \mathbf{A}||_1, \qquad (2)$$

Therefore, the rewired graph topology can be obtained via a constrained optimization problem $\min_{\hat{\mathbf{A}}} \mathcal{L}(\hat{\mathbf{A}}|\mathbf{s}, \mathbf{y}, \mathbf{A}) s.t. \hat{\mathbf{A}}_{ij} \in \{0, 1\}$, where $\alpha$ and $\beta$ are the hyperparameters for label homophily coefficient and graph topology perturbation. Considering the formulated problem is a large-scale discrete optimization problem, we employ a heuristic optimization method to obtain the modified graph topology.

**Optimization Strategy.** To optimize the formulated problem with constraint, we adopt Proximal Gradient Descent (PGD). Specifically, we first adopt gradient descent to update the graph topology using gradient $\frac{\partial \mathcal{L}(\hat{\mathbf{A}}|\mathbf{s}, \mathbf{y}, \mathbf{A})}{\partial \hat{\mathbf{A}}}$. Subsequently, we clip the graph topology $\hat{\mathbf{A}}$ within $\{0, 1\}$ in the projection operation of PGD. Such operation will conduct multiple times to obtain the final graph topology. In practice, considering only the sensitive attribute and label for the training set are available, we actually only modify the graph topology within the training nodes. In other words, the three objectives, including sensitive homophily coefficient, label homophily coefficient, and graph topology perturbation, are calculated for the subgraph of training nodes. In this way, we can avoid information leakage from the test set and reduce the complexity of the optimization problem.

**Evaluation and Computation Complexity Analysis.** For algorithmic evaluation of pre-processing FairGR, we compared the prediction performance (including accuracy and fairness) using original graph topology and rewired graph topology across multiple GNN backbones. Denote the number of training nodes and update iterations as $N$ and $T$, respectively. Then the computation complexity for gradient computation and projection of PGD are both $O(n_{train}^2)$. The total computation complexity to obtain the final rewired graph topology is given by $O(Tn_{train}^2)$. The memory consumption is $O(n_{train}^2)$ due to the storage of graph topology gradient.

Table 1: The performance on Node Classification (GR represents graph topology rewire).

| Models | Pokec-z | | | Pokec-n | | | NBA | | |
|---|---|---|---|---|---|---|---|---|---|
| | Acc (%) ↑ | $\Delta_{DP}$ (%) ↓ | $\Delta_{EO}$ (%) ↓ | Acc (%) ↑ | $\Delta_{DP}$ (%) ↓ | $\Delta_{EO}$ (%) ↓ | Acc (%) ↑ | $\Delta_{DP}$ (%) ↓ | $\Delta_{EO}$ (%) ↓ |
| MLP | 70.48 ± 0.77 | 1.61 ± 1.29 | 2.22 ± 1.01 | 72.48 ± 0.26 | 1.53 ± 0.89 | 3.39 ± 2.37 | 65.56 ± 1.62 | 22.37 ± 1.87 | 18.00 ± 3.52 |
| GAT | 69.76 ± 1.30 | 2.39 ± 0.62 | 2.91 ± 0.97 | 71.00 ± 0.48 | 3.71 ± 2.15 | 7.50 ± 2.88 | 57.78 ± 10.65 | 20.12 ± 16.18 | 13.00 ± 13.37 |
| GAT-GR | 56.75 ± 6.32 | 1.04 ± 0.80 | 1.14 ± 1.02 | 61.27 ± 9.34 | **0.54** ± 0.51 | **2.27** ± 1.55 | 53.65 ± 10.31 | **4.16** ± 5.13 | **3.67** ± 3.23 |
| GCN | 71.78 ± 0.37 | 3.25 ± 2.35 | 2.36 ± 2.09 | 73.09 ± 0.28 | 3.48 ± 0.47 | 5.16 ± 1.38 | 61.90 ± 1.00 | 23.70 ± 2.74 | 17.50 ± 2.63 |
| GCN-GR | 71.68 ± 0.58 | 1.94 ± 1.59 | **1.27** ± 0.71 | 72.68 ± 0.44 | **0.47** ± 0.39 | **0.82** ± 0.78 | 61.59 ± 1.85 | **20.24** ± 4.41 | **9.50** ± 2.77 |
| SGC | 71.24 ± 0.46 | 4.81 ± 0.30 | 4.79 ± 2.27 | 71.46 ± 0.41 | 2.22 ± 0.29 | 3.85 ± 1.63 | 63.17 ± 0.63 | 22.56 ± 3.94 | 14.33 ± 2.16 |
| SGC-GR | 70.95 ± 0.91 | 3.32 ± 1.31 | 3.20 ± 1.90 | 71.91 ± 0.52 | **0.71** ± 0.65 | 2.39 ± 0.69 | 62.54 ± 1.62 | **18.56** ± 2.81 | **2.50** ± 1.66 |

## 6 EXPERIMENTS

In this section, we conduct experiments to validate the effectiveness (see Apppendix G for more details.) of the proposed FairGR. We firstly validate that GCN-like aggregation enhances representation bias for the graph data with large sensitive homophily via synthetic experiments. For real-world datasets, we conduct experiments to show that the prediction bias of GNN is larger than that of MLP. Moreover, we introduce the experimental settings and then evaluate our proposed FairGR compared with several baselines in terms of prediction performance and fairness metrics on real-world datasets.

### 6.1 SYNTHETIC EXPERIMENTS

In the synthetic experiments, we demonstrate the relation between DP difference across GCN-like message passing operation and sensitive homophily coefficient [2]. Specifically, we investigate the

---

[2] Note that we only do a theoretical study in GCN-like message passing operation as a pilot study. The investigation of other GNN aggregation operations (such as GraghSAGE-like operation) and GNN models may require different techniques and can be further conducted in future work.

influence of graph node number $n$, graph connection density $\rho_d$, sensitive homophily $\epsilon_{sens}$, and sensitive attribute ration $c$ for bias enhancement of GCN-like message passing. **For evaluation metric,** we adopt the demographic parity (DP) difference during message passing to measure the bias enhancement. For **node attribute generation**, we first generate node attribute with Gaussian distribution $\mathcal{N}(\mu_1, \boldsymbol{\Sigma})$ and $\mathcal{N}(\mu_2, \boldsymbol{\Sigma})$ for node with binary sensitive attribute, respectively, where $\mu_1 = [0, 1]$, $\mu_1 = [1, 0]$ and $\boldsymbol{\Sigma} = \begin{bmatrix} 1 & 0 \\ 0 & 1 \end{bmatrix}$. **For adjacency matrix generation**, we randomly generate edges via a stochastic block model based on the intra-connection and inter-connection probability.

Figure 2 shows the DP difference during message passing with respect to the sensitive homophily coefficient. We observe that a higher sensitive homophily coefficient generally leads to larger bias enhancement. Additionally, higher graph connection density $\rho_d$, larger node number $n$, and balanced sensitive attribute ratio $c$ correspond to higher bias enhancement, which is consistent with our theoretical analysis in Theorem 2.

### 6.2 EXPERIMENTS ON REAL-WORLD DATASETS

#### 6.2.1 EXPERIMENTAL SETTINGS

**Datasets.** The experiments are conducted on three real-world datasets, including Pokec-z, Pokec-n, and NBA (Dai & Wang, 2021). Pokec-z and Pokec-n are sampled from a larger social network Pokec (Takac & Zabovsky, 2012) based on the province in Slovakia. We choose region information and the working field of the users as the sensitive attribute and the predicted label, respectively. NBA dataset includes around 400 NBA basketball players and is collected from a Kaggle dataset [3] and Twitter. The information of players includes age, nationality, and salary in the 2016-2017 season. We choose nationality (U.S. and overseas player) as the binary sensitive attribute, and the prediction label is whether the salary is higher than the median.

**Evaluation Metrics.** For the node classification task, we adopt accuracy to evaluate the classification performance. As for fairness metric, we adopt two most common-used group fairness metrics, including *demographic parity* and *equal opportunity*, to measure the prediction bias (Louizos et al., 2015; Beutel et al., 2017). Specifically, *demographic parity* is defined as the average prediction difference over different sensitive attribute groups, i.e., $\Delta_{DP} = |\mathbb{P}(\hat{y} = 1|s = -1) - \mathbb{P}(\hat{y} = 1|s = 1)|$. Similarly, *equal opportunity* is given by $\Delta_{EO} = |\mathbb{P}(\hat{y} = 1|s = -1, y = 1) - \mathbb{P}(\hat{y} = 1|s = 1, y = 1)|$, where $y$ and $\hat{y}$ represent the ground-truth label and predicted label, respectively.

**Baselines.** Considering that the proposed FairGR is a pre-processing method, we show that our proposed FairGR can improve many representative GNNs, such as GCN (Kipf & Welling, 2017), GAT (Veličković et al., 2018), SGC (Wu et al., 2019), in many fairness training strategies, such adding regularization, adversarial debiasing. For all models, we train 2 layers of neural networks with 64 hidden units for 300 epochs.

**Implementation Details.** For each experiment, we run 5 times and report the average performance for each method. We adopt Adam optimizer with $0.001$ learning rate and $1e^{-5}$ weight decay for all models training. In adversarial debiasing setting, we train the classifier and adversary head with 70 and 30 epochs, respectively. The hyperparameters for adversary debiasing are tuned in $\{0.0, 0.5, 1.0, 2.0, 5.0, 8.0, 10.0, 50.0, 100.0\}$. For adding regularization, we adopt the hyperparameter set $\{0.0, 1.0, 1.5, 2.0, 5.0, 8.0, 10.0, 15.0, 25.0, 50.0, 80.0, 100.0\}$.

#### 6.2.2 DOES GNNs HAVE A LARGER PREDICTION BIAS THAN MLP?

To validate the effect of bias enhancement of GNNs, we compare the performance of many representative GNNs over MLP on various real-world datasets and summarize the results in Table 1. From these results, we make the following observations:

- Many representative GNNs have a higher prediction bias compared with MLP model on all three datasets in terms of demographic parity and equal opportunities. For demographic parity, the

---

[3]https://www.kaggle.com/noahgift/social-power-nba

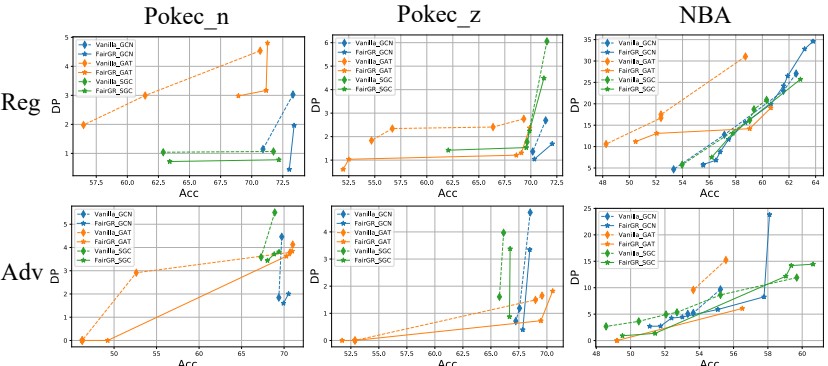

Figure 3: DP and Acc trade-off performance on three real-world datasets compared with adding regularization (Top) and adversarial debiasing (Bottom). The trade-off curve closer to the right bottom corner represents better trade-off performance.

prediction bias of MLP is lower than that of GAT, GCN, and SGC by $32.64\%$, $50.46\%$, $66.53\%$ and $58.72\%$ on Pokec-z dataset. The higher prediction bias comes from the aggregation within the same-sensitive attribute nodes and topology bias in graph data.

- FairGR can mitigate bias for GCN and SGC backbone via rewiring graph topology in these three datasets. For GAT backbone, although the bias can be mitigated, the accuracy drop is significant due to the fact that GAT is more sensitive to graph topology rewire.

### 6.2.3 DOES FAIRGR ACHIEVE BETTER TRADEOFF PERFORMANCE IN VARIOUS SETTINGS?

**Comparison with adversarial debiasing and regularization.** To validate that the proposed FairGR is compatible with many fairness training strategies, we also show the prediction performance and fairness metric trade-off compared with adversarial debiasing (Fisher et al., 2020) and add demographic parity regularization (Chuang & Mroueh, 2020). In adversarial debiasing (Louppe et al., 2017), the output of GNNs is the input of the adversary, and the goal of the adversary is to predict the node sensitive attribute. For these two fair training strategies, we adopt GCN, GAT, and SGC as backbones. We randomly split $50\%/25\%/25\%$ for training, validation, and test dataset. Figure 3 shows the Pareto optimality curve for all methods, where the right-bottom corner point represents the ideal performance ($100\%$ accuracy and $0\%$ prediction bias). From the results, we list the following observations as follows:

- For both adversarial debiasing and adding regularization training strategies, our proposed FairGR can achieve a better DP-Acc trade-off compared with that without any graph data refinement for many GNNs. In other words, FairGR can effectively reduce training bias and is compatible with many existing fairness training strategies.

- Topology does matter in GNNs. For adding regularization or adversarial debiasing, FairGS embrace different tradeoff performance gain on top of different GNNs. Such observation implies that there is a complicated interaction between graph topology and message passing algorithms. Additionally, FairGS provide the most tradeoff performance benefit in GAT compared with GCN and SGC. The high capacity of GAT may energize the message passing algorithm to learn from data. Therefore, the tradeoff performance improvement is the highest in adding regularization and adversarial debiasing.

## 7 CONCLUSION

In this work, we theoretically demonstrate that the message passing amplifies node representation bias under the graph data with a large sensitive homophily coefficient, and reveal the role of other graphs statistical information in terms of bias amplification. Additionally, motivated by theoretical understanding, we develop a simple yet effective graph refinement method, named FairGR, to reduce the sensitive homophily while preserving useful information. We conduct synthetic experiments to validate theoretical findings. Experimental results on real-world datasets demonstrate the effectiveness of FairGR in many fair training strategies and GNNs backbones in node classification tasks.

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

## A    PROOF OF THEOREM 1

We provide a more general proof for categorical sensitive attribute $\mathbf{s} \in \{1, 2, \cdots, K\}$ and the prior probability is given by $\mathbb{P}(\mathbf{s} = i) = c_i$. Suppose the conditional node attribute $\mathbf{x}$ distribution given node sensitive attribute $\mathbf{s} = i$ satisfies normal distribution $P_i(\mathbf{x}) \overset{\triangle}{=} \mathcal{N}(\mu_i, \mathbf{\Sigma_i})$, the distribution of node sensitive attribute is the mixed Gaussian distribution $f(\mathbf{x}) = \sum_{i=1}^{K} c_i P_i(\mathbf{x})$. Based on the definition of mutual information, we have

$$I(\mathbf{s}, \mathbf{x}) = H(\mathbf{x}) - \sum_{i=1}^{K} c_i H(\mathbf{x}|\mathbf{s} = i); \tag{3}$$

where $H(\cdot)$ represents Shannon entropy for random variable. Subsequently, we focus on the entropy of the mixed Gaussian distribution $H(\mathbf{x})$. We show that such entropy can be upper bounded by the pairwise Kullback-Leibler (KL) divergence as follows:

$$
\begin{aligned}
I(\mathbf{s}, \mathbf{x}) &= -\sum_{i=1}^{K} c_i \mathbb{E}_{P_i}\big[\ln \sum_{j=1}^{K} c_j P_j(\mathbf{x})\big] - \sum_{i=1}^{K} c_i H(\mathbf{x}|\mathbf{s} = i) \\
&\overset{(a)}{\leq} -\sum_{i=1}^{K} c_i \big[\ln \sum_{j=1}^{K} c_j e^{\mathbb{E}_{P_i}[\ln P_j(\mathbf{x})]}\big] - \sum_{i=1}^{K} c_i H(\mathbf{x}|\mathbf{s} = i) \\
&= -\sum_{i=1}^{K} c_i \big[\ln \sum_{j=1}^{K} c_j e^{-H(P_i||P_j)}\big] - \sum_{i=1}^{K} c_i H(\mathbf{x}|\mathbf{s} = i) \\
&\overset{(b)}{=} -\sum_{i=1}^{K} c_i \big[\ln \sum_{j=1}^{K} c_j e^{-H(P_i) - D_{KL}(P_i||P_j)}\big] - \sum_{i=1}^{K} c_i H(\mathbf{x}|\mathbf{s} = i) \\
&= -\sum_{i=1}^{K} c_i \big[\ln \sum_{j=1}^{K} c_j e^{-D_{KL}(P_i||P_j)}\big] + \sum_{i=1}^{K} c_i H(P_i) - \sum_{i=1}^{K} c_i H(\mathbf{x}|\mathbf{s} = i), \\
&= -\sum_{i=1}^{K} c_i \big[\ln \sum_{j=1}^{K} c_j e^{-D_{KL}(P_i||P_j)}\big],
\end{aligned}
$$

where KL divergence $D_{KL}(P_i||P_j) \overset{\triangle}{=} \int P_i(\mathbf{x}) \ln \frac{P_i(\mathbf{x})}{P_j(\mathbf{x})} d\mathbf{x}$ and cross entropy $H(P_i||P_j) = -\int P_i(\mathbf{x}) \ln P_j(\mathbf{x}) d\mathbf{x}$. The inequality (a) holds based on the variational lower bound on the expectation of a log-sum inequality $\mathbb{E}\big[\ln \sum_i Z_i\big] \geq \ln \big[\sum_i e^{\mathbb{E}[\ln Z_i]}\big]$ (Kullback, 1997), and quality (2) holds based on $H(P_i||P_j) = H(P_i) + D_{KL}(P_i||P_j)$. As a special case for binary sensitive attribute, it is easy to obtain the following results:

$$I(\mathbf{s}, \mathbf{X}) \leq -(1-c)\ln\big[(1-c) + c\exp\big(-D_{KL}(P_1||P_2)\big)\big] - c\ln\big[c + (1-c)\exp\big(-D_{KL}(P_2||P_1)\big)\big].$$

## B    PROOF OF THEOREM 2

Before going deeper for our proof, we first introduce two useful lemmas on KL divergence and statistical information of graph.

**Lemma 2** *For two d-dimensional Gaussian distributions $P = \mathcal{N}(\mu_p, \mathbf{\Sigma}_p)$ and $Q = \mathcal{N}(\mu_q, \mathbf{\Sigma}_q)$, the KL divergence $D_{KL}(P||Q)$ is given by*

$$D_{KL}(P||Q) = \frac{1}{2}\Big[\ln \frac{|\mathbf{\Sigma}_q|}{|\mathbf{\Sigma}_p|} - d + (\mu_p - \mu_q)^\top \mathbf{\Sigma}_q^{-1}(\mu_p - \mu_q) + Tr(\mathbf{\Sigma}_q^{-1}\mathbf{\Sigma}_p)\Big] \tag{4}$$

*where $\top$ is matrix transpose operation and $Tr(\cdot)$ is trace of a square matrix.*

**Proof:** *Note that the probability density function of multivariate Normal distribution is given by:*

$$P(\mathbf{x}) = \frac{1}{(2\pi)^{d/2}|\mathbf{\Sigma}_p|^{1/2}} \exp\Big(-\frac{1}{2}(\mathbf{x}-\mu_p)^\top \mathbf{\Sigma}_p^{-1}(\mathbf{x}-\mu_p)\Big),$$

*the KL divergence between distributions $P$ and $Q$ can be given by*

$$
\begin{aligned}
D_{KL}(P||Q) &= \mathbb{E}_P[\ln(P) - \ln(Q)] \\
&= \mathbb{E}_P\Big[\frac{1}{2}\ln\frac{|\mathbf{\Sigma}_q|}{|\mathbf{\Sigma}_p|} - \frac{1}{2}(\mathbf{x}-\mu_p)^\top\mathbf{\Sigma}_p^{-1}(\mathbf{x}-\mu_p) + \frac{1}{2}(\mathbf{x}-\mu_q)^\top\mathbf{\Sigma}_q^{-1}(\mathbf{x}-\mu_q)\Big] \\
&= \frac{1}{2}\ln\frac{|\mathbf{\Sigma}_q|}{|\mathbf{\Sigma}_p|} - \underbrace{\frac{1}{2}\mathbb{E}_P\Big[(\mathbf{x}-\mu_p)^\top\mathbf{\Sigma}_p^{-1}(\mathbf{x}-\mu_p)\Big]}_{I_1} + \underbrace{\frac{1}{2}\mathbb{E}_P\Big[(\mathbf{x}-\mu_q)^\top\mathbf{\Sigma}_q^{-1}(\mathbf{x}-\mu_q)\Big]}_{I_2}.
\end{aligned}
$$

*Using the commutative property of the trace operation, we have*

$$
\begin{aligned}
I_1 &= \frac{1}{2}\mathbb{E}_P\Big[(\mathbf{x}-\mu_p)^\top\mathbf{\Sigma}_p^{-1}(\mathbf{x}-\mu_p)\Big] = \frac{1}{2}Tr\Big(\mathbb{E}_P\Big[(\mathbf{x}-\mu_p)^\top(\mathbf{x}-\mu_p)\mathbf{\Sigma}_p^{-1}\Big]\Big) \\
&= \frac{1}{2}Tr\Big(\mathbb{E}_P\Big[(\mathbf{x}-\mu_p)^\top(\mathbf{x}-\mu_p)\Big]\mathbf{\Sigma}_p^{-1}\Big) = \frac{1}{2}Tr\Big(\mathbf{\Sigma}_p\mathbf{\Sigma}_p^{-1}\Big) = \frac{d}{2},
\end{aligned}
\tag{5}
$$

*As for the term $I_2$, note that $\mathbf{x} - \mu_q = (\mathbf{x} - \mathbb{E}_P[\mathbf{x}]) + (\mathbb{E}_P[\mathbf{x}] - \mu_q)$, we can obtain the following equation:*

$$
\begin{aligned}
I_2 &= \frac{1}{2}\mathbb{E}_P\Big[(\mathbf{x}-\mu_q)^\top\mathbf{\Sigma}_q^{-1}(\mathbf{x}-\mu_q)\Big] \\
&= \frac{1}{2}\Big[(\mu_p-\mu_q)^\top\mathbf{\Sigma}_q^{-1}(\mu_p-\mu_q) + Tr(\mathbf{\Sigma}_q^{-1}\mathbf{\Sigma}_p)\Big],
\end{aligned}
\tag{6}
$$

*Therefore, the KL divergence $D_{KL}(P||Q)$ is given by*

$$D_{KL}(P||Q) = \frac{1}{2}\Big[\ln\frac{|\mathbf{\Sigma}_q|}{|\mathbf{\Sigma}_p|} - d + (\mu_p-\mu_q)^\top\mathbf{\Sigma}_q^{-1}(\mu_p-\mu_q) + Tr(\mathbf{\Sigma}_q^{-1}\mathbf{\Sigma}_p)\Big]. \tag{7}$$

$\square$

**Lemma 3** *Suppose the synthetic graph is generated from $(n, \rho_d, \epsilon_{sens}, c)$-graph, then we obtain the intra-connect and inter-connect probability as follows:*

$$p_{conn} = \frac{\rho_d\epsilon_{sens}}{c^2 + (1-c)^2}, \quad q_{conn} = \frac{\rho_d(1-\epsilon_{sens})}{2c(1-c)}.$$

**Proof:** *Based on Bayes' rule, we have the intra-connect and inter-connect probability as follows*

$$p_{conn} = \mathbb{P}(\mathbf{A}_{ij}=1|\mathbf{s}_i=\mathbf{s}_j) = \frac{\mathbb{P}(\mathbf{A}_{ij}=1)\mathbb{P}(\mathbf{s}_i=\mathbf{s}_j|\mathbf{A}_{ij}=1)}{\mathbb{P}(\mathbf{s}_i=\mathbf{s}_j)} = \frac{\rho_d\epsilon_{sens}}{c^2+(1-c)^2},$$

$$q_{conn} = \mathbb{P}(\mathbf{A}_{ij}=1|\mathbf{s}_i\mathbf{s}_j=-1) = \frac{\mathbb{P}(\mathbf{A}_{ij}=1)\mathbb{P}(\mathbf{s}_i\mathbf{s}_j=-1|\mathbf{A}_{ij}=1)}{\mathbb{P}(\mathbf{s}_i\mathbf{s}_j)=-1} = \frac{\rho_d(1-\epsilon_{sens})}{2c(1-c)}. \tag{8}$$

$\square$

Note that the synthetic graph is generated from $(n, \rho_d, \epsilon_{sens}, c)$-graph. The sensitive attribute $\mathbf{s}$ is generated to with ratio $c$, i.e., the number of node sensitive attribute $\mathbf{s} = -1$ and $\mathbf{s} = 1$ are $n_{-1} = n(1-c)$ and $n_1 = nc$. Based on the determined sensitive attribute $\mathbf{s}$, we randomly generate the edge based on parameters $\rho_d$ and $\epsilon_{sens}$ and Lemma 1, i.e., the edges within and cross the same group are randomly generated based on intra-connect probability and inter-connect probability. Therefore, the adjacency matrix $\mathbf{A}_{ij}$ is independent on node attribute $\mathbf{X}_i$ and $\mathbf{X}_j$ given sensitive attributes $\mathbf{s}_i$ and $\mathbf{s}_j$, i.e., $\mathbf{A}_{ij} \perp\!\!\!\perp (\mathbf{X}_i, \mathbf{X}_j)|(\mathbf{s}_i, \mathbf{s}_j)$. Similarly, the different node attributes and edges are also dependent on each other given sensitive attributes, i.e., $\mathbf{A}_{ij} \perp\!\!\!\perp \mathbf{A}_{ij}|(\mathbf{s}_i, \mathbf{s}_j, \mathbf{s}_k)$ and

$\mathbf{X}_i \perp\!\!\!\perp \mathbf{X}_j | (\mathbf{s}_i, \mathbf{s}_j)$. Therefore, considering GCN-like message passing $\tilde{\mathbf{X}}_i = \sum_{j=1}^n \tilde{\mathbf{A}}_{ij} \mathbf{X}_j$, we have the aggregated node attributes expectation given sensitive attribute as follows:

$$
\begin{aligned}
\tilde{\mu}_1 &= \mathbb{E}_{\tilde{\mathbf{X}}_i}[\tilde{\mathbf{X}}_i | \mathbf{s}_i = -1] = \sum_{j=1}^n \mathbb{E}_{\tilde{\mathbf{A}}_{ij}, \mathbf{X}_j}[\tilde{\mathbf{A}}_{ij} \mathbf{X}_j | \mathbf{s}_i = -1] \\
&= \sum_{j=1}^n \mathbb{E}_{\tilde{\mathbf{A}}_{ij}}[\tilde{\mathbf{A}}_{ij} | \mathbf{s}_i = -1] \mathbf{E}_{\mathbf{X}_j}[\mathbf{X}_j | \mathbf{s}_i = -1] \\
&= (n_{-1} - 1)\mathbb{E}_{\tilde{\mathbf{A}}_{ij}}[\tilde{\mathbf{A}}_{ij} | \mathbf{s}_i = -1, \mathbf{s}_j = -1] \mathbf{E}_{\mathbf{X}_j}[\tilde{\mathbf{X}}_j | \mathbf{s}_i = -1, \mathbf{s}_j = -1] \\
&\quad + \mathbb{E}_{\tilde{\mathbf{A}}_{ij}}[\tilde{\mathbf{A}}_{ij} | \mathbf{s}_i = -1, i = j] \mathbf{E}_{\mathbf{X}_j}[\mathbf{X}_j | \mathbf{s}_i = -1] \\
&\quad + n_1 \mathbb{E}_{\tilde{\mathbf{A}}_{ij}}[\tilde{\mathbf{A}}_{ij} | \mathbf{s}_i = -1, \mathbf{s}_j = 1] \mathbf{E}_{\mathbf{X}_j}[\mathbf{X}_j | \mathbf{s}_i = -1, \mathbf{s}_j = 1] \\
&= \frac{[(n_{-1} - 1)p_{conn} + 1]\mu_1 + n_1 q_{conn} \mu_2}{(n_{-1} - 1)p_{conn} + 1 + n_1 q_{conn}} \triangleq \nu_1 \mu_1 + (1 - \nu_1)\mu_2. \quad (9)
\end{aligned}
$$

where $\nu_1 = \frac{(n_{-1} - 1)p_{conn} + 1}{(n_{-1} - 1)p_{conn} + 1 + n_1 q_{conn}}$. Similarly, for the node with sensitive attribute 1, we have

$$
\begin{aligned}
\tilde{\mu}_2 &= \mathbb{E}_{\tilde{\mathbf{X}}_i}[\tilde{\mathbf{X}}_i | \mathbf{s}_i = 1] = \sum_{j=1}^n \mathbb{E}_{\tilde{\mathbf{A}}_{ij}, \mathbf{X}_j}[\tilde{\mathbf{A}}_{ij} \mathbf{X}_j | \mathbf{s}_i = 1] \\
&= \sum_{j=1}^n \mathbb{E}_{\tilde{\mathbf{A}}_{ij}}[\tilde{\mathbf{A}}_{ij} | \mathbf{s}_i = 1] \mathbf{E}_{\mathbf{X}_j}[\mathbf{X}_j | \mathbf{s}_i = 1] \\
&= n_{-1} \mathbb{E}_{\tilde{\mathbf{A}}_{ij}}[\tilde{\mathbf{A}}_{ij} | \mathbf{s}_i = 1, \mathbf{s}_j = -1] \mathbf{E}_{\mathbf{X}_j}[\mathbf{X}_j | \mathbf{s}_i = 1, \mathbf{s}_j = -1] \\
&\quad + \mathbb{E}_{\tilde{\mathbf{A}}_{ij}}[\tilde{\mathbf{A}}_{ij} | \mathbf{s}_i = 1, i = j] \mathbf{E}_{\mathbf{X}_j}[\mathbf{X}_j | \mathbf{s}_i = 1] \\
&\quad + (n_1 - 1)\mathbb{E}_{\tilde{\mathbf{A}}_{ij}}[\tilde{\mathbf{A}}_{ij} | \mathbf{s}_i = 1, \mathbf{s}_j = 1] \mathbf{E}_{\mathbf{X}_j}[\mathbf{X}_j | \mathbf{s}_i = 1, \mathbf{s}_j = 1] \\
&= \frac{n_{-1} q_{conn} \mu_1 + [(n_1 - 1)p_{conn} + 1]\mu_2}{n_{-1} q_{conn} + (n_1 - 1)p_{conn} + 1} \triangleq \nu_2 \mu_1 + (1 - \nu_2)\mu_2. \quad (10)
\end{aligned}
$$

where $\nu_2 = \frac{(n_{-1} - 1)q_{conn}}{n_{-1} q_{conn} + 1 + (n_1 - 1)p_{conn}}$. As for the covariance matrix of aggregated node attributes $\tilde{\mathbf{X}}$ given node sensitive attribute $\mathbf{s} = -1$ and original sensitive attribute, note that we can obtain

$$
\begin{aligned}
\tilde{\boldsymbol{\Sigma}}_1 &= \mathbb{D}_{\tilde{\mathbf{X}}_i}[\tilde{\mathbf{X}}_i | \mathbf{s}_i = -1] = \sum_{j=1}^n \mathbb{D}_{\tilde{\mathbf{A}}_{ij}, \mathbf{X}_j}[\tilde{\mathbf{A}}_{ij} \mathbf{X}_j | \mathbf{s}_i = -1] \\
&= \sum_{j=1}^n \mathbb{E}_{\tilde{\mathbf{A}}_{ij}}[\tilde{\mathbf{A}}_{ij}^2 | \mathbf{s}_i = -1] \mathbf{D}_{\tilde{\mathbf{X}}_j}[\tilde{\mathbf{X}}_j | \mathbf{s}_i = -1] \\
&= \frac{(n_{-1} - 1)p_{conn}\boldsymbol{\Sigma} + \boldsymbol{\Sigma} + n_1 q_{conn}\boldsymbol{\Sigma}}{[(n_{-1} - 1)p_{conn} + 1 + n_1 q_{conn}]^2} \\
&= \frac{\boldsymbol{\Sigma}}{(n_{-1} - 1)p_{conn} + 1 + n_1 q_{conn}} \triangleq \zeta_1^{-1}\boldsymbol{\Sigma}, \quad (11)
\end{aligned}
$$

where $\zeta_1 = (n_{-1} - 1)p_{conn} + 1 + n_1 q_{conn}$. Similarly, given node sensitive attribute $\mathbf{s} = 1$, we have $\tilde{\boldsymbol{\Sigma}}_2 = \mathbb{D}_{\tilde{\mathbf{X}}_i}[\tilde{\mathbf{X}}_i | \mathbf{s}_i = 1] = \frac{\boldsymbol{\Sigma}}{n_{-1} q_{conn} + 1 + (n_1 - 1)p_{conn}} \triangleq \zeta_2^{-1}\boldsymbol{\Sigma}$, where $\zeta_2 = n_{-1} q_{conn} + 1 + (n_1 - 1)p_{conn}$. In other words, the covariance matrix of the aggregated node attributes is lower than the original one since the "average" operation will make node representation more concentrated. Note that the summation over several Gaussian random variables is still Gaussian, we define the node attributes distribution for sensitive attribute $\mathbf{s} = -1$ and $\mathbf{s} = 1$ as $P_1 = \mathcal{N}(\mu_1, \boldsymbol{\Sigma})$, $P_2 = \mathcal{N}(\mu_2, \boldsymbol{\Sigma})$, respectively. Similarly, the aggregated node representation distribution follows for sensitive attribute $\mathbf{s} = -1$ and $\mathbf{s} = 1$ as $\tilde{P}_1 = \mathcal{N}(\tilde{\mu}_1, \tilde{\boldsymbol{\Sigma}}_1)$, $\tilde{P}_2 = \mathcal{N}(\tilde{\mu}_2, \tilde{\boldsymbol{\Sigma}}_2)$. Note that the sensitive attribute ratio keeps the same after the aggregation and larger KL divergence for these two sensitive attributes group distribution, the bias enhances $\Delta Bias > 0$ if $D_{KL}(\tilde{P}_1 || \tilde{P}_2) > D_{KL}(P_1 || P_2)$ and

$D_{KL}(\tilde{P}_2||\tilde{P}_1) > D_{KL}(P_2||P_1)$. Therefore, we only focus on the KL divergence. According to Lemma 2, it is easy to obtain KL divergence for original distribution as follows:

$$
\begin{aligned}
D_{KL}(P_1||P_2) &= \frac{1}{2}\Big[\ln\frac{|\boldsymbol{\Sigma}|}{|\boldsymbol{\Sigma}|} - d + (\mu_1 - \mu_2)^\top\boldsymbol{\Sigma}^{-1}(\mu_1 - \mu_2) + Tr(\boldsymbol{\Sigma}^{-1}\boldsymbol{\Sigma})\Big] \\
&= \frac{1}{2}(\mu_1 - \mu_2)^\top\boldsymbol{\Sigma}^{-1}(\mu_1 - \mu_2),
\end{aligned}
\tag{12}
$$

As for KL divergence for aggregated distribution, similarly, we have

$$
\begin{aligned}
D_{KL}(\tilde{P}_1||\tilde{P}_2) &= \frac{1}{2}\Big[\ln\frac{|\tilde{\boldsymbol{\Sigma}}_2|}{|\tilde{\boldsymbol{\Sigma}}_1|} - d + (\tilde{\mu}_1 - \tilde{\mu}_2)^\top\tilde{\boldsymbol{\Sigma}}_2^{-1}(\tilde{\mu}_1 - \tilde{\mu}_2) + Tr(\tilde{\boldsymbol{\Sigma}}_2^{-1}\tilde{\boldsymbol{\Sigma}}_1)\Big] \\
&= \frac{1}{2}\Big[d\ln\frac{\zeta_1}{\zeta_2} - d + (\nu_1 - \nu_2)^2\zeta_2(\mu_1 - \mu_2)^\top\boldsymbol{\Sigma}^{-1}(\mu_1 - \mu_2) + \frac{\zeta_2}{\zeta_1}Tr(\mathbf{I}_d)\Big] \\
&\overset{(c)}{\geq} \frac{1}{2}(\nu_1 - \nu_2)^2\zeta_2(\mu_1 - \mu_2)^\top\boldsymbol{\Sigma}^{-1}(\mu_1 - \mu_2),
\end{aligned}
\tag{13}
$$

where inequality $(c)$ holds since $\ln x \leq x - 1$ for any $x > 0$. Compared with equations (12) and (13), it is seen that $D_{KL}(\tilde{P}_1||\tilde{P}_2) > D_{KL}(P_1||P_2)$ if $(\nu_1 - \nu_2)^2\zeta_2 > 1$. Similarly, we can have $D_{KL}(\tilde{P}_2||\tilde{P}_1) > D_{KL}(P_2||P_1)$ if $(\nu_1 - \nu_2)^2\zeta_1 > 1$. In a nutshell, the bias enhances $\Delta Bias > 0$ after message passing if $(\nu_1 - \nu_2)^2\min\{\zeta_1, \zeta_2\} > 1$.

## C  Topology Amplifies Bias in One-Layer GCN

Section 4 demonstrates that GCN-like aggregation operation amplifies node representation bias for graph data with large topology bias. However, it is still unclear whether such observation holds for general GNNs or not. Generally speaking, this problem is quite fundamental and challenging to understand the role of topology in fair graph learning. In this section, we try to move a step toward this problem by considering one-layer GCN. Prior to comparing the prediction for one-layer GCN and one-layer MLP, we first provide the connection between demographic parity and mutual information of sensitive attributes and predictions. Then, we theoretically compare the prediction bias of one-layer GCN and one-layer MLP through the lens of mutual information.

### C.1  Prediction Bias and Mutual Information

Here, we only consider binary sensitive attributes $\mathbf{s} \in \{-1, 1\}$ and binary labels $\mathbf{y} \in \{-1, 1\}$. Similarly, we can define $\hat{\mathbf{y}} \in \{-1, 1\}$ as the binary model predictions. In the fairness community, demographic parity, defined as the average prediction difference among different sensitive attribute groups, is the most commonly used fairness metric, i.e., $\Delta DP = |\mathbb{P}(\hat{\mathbf{y}}|\mathbf{s} = 1) - \mathbb{P}(\hat{\mathbf{y}}|\mathbf{s} = -1)|$. From the mutual information perspective, the correlation between sensitive attributes $\mathbf{s}$ and prediction $\hat{\mathbf{y}}$ can be measured by $I(\mathbf{s}; \hat{\mathbf{y}})$. In this subsection, we provide an inherent connection between demographic parity $\Delta DP$ and mutual information $I(\mathbf{s}; \hat{\mathbf{y}})$ as follows:

**Theorem 3** *For binary sensitive attributes $\mathbf{s} \in \{-1, 1\}$ and binary prediction $\hat{\mathbf{y}} \in \{-1, 1\}$, demographic parity $\Delta DP$ and mutual information $I(\mathbf{s}; \hat{\mathbf{y}})$ satisfies $I(\mathbf{s}; \hat{\mathbf{y}}) \leq 2\Delta DP$*

**Proof:**  *For simplicity, we defined the joint probability as $\alpha_{i\cup j} = \mathbb{P}(\hat{\mathbf{y}} = i, \mathbf{s} = j)$ and condition probability as $\alpha_{i|j} = \mathbb{P}(\hat{\mathbf{y}} = i|\mathbf{s} = j)$. Considering the log ratio between joint distribution and margin product probability, we have*

$$
\begin{aligned}
\log\frac{\mathbb{P}(\hat{\mathbf{y}} = i, \mathbf{s} = j)}{\mathbb{P}(\hat{\mathbf{y}} = i)\mathbb{P}(\mathbf{s} = j)} &= \log\frac{\alpha_{i|j}}{\sum_j \alpha_{i|j}\mathbb{P}(\mathbf{s} = j)} = \log\Big(1 + \frac{(\alpha_{i|j} - \alpha_{i|-j})\mathbb{P}(\mathbf{s} = -j)}{\sum_j \alpha_{i|j}\mathbb{P}(\mathbf{s} = j)}\Big) \\
&\overset{(d)}{\leq} (\alpha_{i|j} - \alpha_{i|-j})\frac{\mathbb{P}(\mathbf{s} = -j)}{\sum_j \alpha_{i|j}\mathbb{P}(\mathbf{s} = j)} \leq \Delta DP\frac{\mathbb{P}(\mathbf{s} = -j)}{\sum_j \alpha_{i|j}\mathbb{P}(\mathbf{s} = j)}.
\end{aligned}
\tag{14}
$$

*where inequality (d) holds due to $\log(1+x) \leq x$ for any $x > -1$. According to the definition of mutual information, we have*

$$
\begin{aligned}
I(\mathbf{s}; \hat{\mathbf{y}}) &= \sum_{i,j} \mathbb{P}(\hat{\mathbf{y}}=i, \mathbf{s}=j) \log \frac{\mathbb{P}(\hat{\mathbf{y}}=i, \mathbf{s}=j)}{\mathbb{P}(\hat{\mathbf{y}}=i)\mathbb{P}(\mathbf{s}=j)} = \sum_{i,j} \alpha_{i \cup j} \log \frac{\alpha_{i|j}}{\sum_j \alpha_{i|j}\mathbb{P}(\mathbf{s}=j)} \\
&\leq \Delta DP \sum_{i,j} \alpha_{i \cup j} \frac{\mathbb{P}(\mathbf{s}=-j)}{\sum_j \alpha_{i|j}\mathbb{P}(\mathbf{s}=j)} = \Delta DP \sum_{i,j} \mathbb{P}(\mathbf{s}=-j)\mathbb{P}(\mathbf{s}=j)|\hat{\mathbf{y}}=i) \\
&\overset{(f)}{\leq} \Delta DP \sum_i \Big[ \sum_j \mathbb{P}(\mathbf{s}=-j) \Big] \Big[ \sum_j \mathbb{P}(\mathbf{s}=j|\hat{\mathbf{y}}=i) \Big] = 2\Delta DP
\end{aligned}
\tag{15}
$$

*where inequality (f) holds due to $\sum_i a_i b_i \leq \sum_i a_i \sum_i b_i$ for non-negative $a_i$ and $b_i$.* □

Theorem 3 shows there is a strong connection between demographic parity and mutual information for binary sensitive attributes and binary labels, i.e., the mutual information is upper bounded by demographic parity multiplied by 2.

## C.2 PREDICTION BIAS COMPARISON BETWEEN ONE-LAYER GCN AND ONE-LAYER MLP

Considering the strong connection between mutual information and demographic parity, we investigate prediction bias comparison between one-layer GCN and one-layer MLP through the lens of mutual information. For one-layer MLP model, the prediction is given by $\hat{\mathbf{y}}_{MLP} = \sigma(\mathbf{X}\mathbf{W}_{MLP})$, where $\mathbf{W}_{MLP}$ is trainable parameter for MLP. Similarly, for one-layer GCN, the prediction is given by $\hat{\mathbf{y}}_{GCN} = \sigma(\tilde{\mathbf{A}}\mathbf{X}\mathbf{W}_{GCN})$, where $\mathbf{W}_{GCN}$ is the trainable parameters for GCN. Define $\tilde{\mathbf{X}} = \tilde{\mathbf{A}}\mathbf{X}$, it is easy to see that one-layer MLP model and one-layer GCN model are almost the same except with different node features. Based on Theorem 2, the aggregated node features $\tilde{\mathbf{X}}$ embrace higher presentation bias than that of $\mathbf{X}$. In other words, the bias of input data for one-layer GCN is higher than that of one-layer MLP.

For the prediction bias, note that sensitive attributes $\mathbf{s}$, node features $\mathbf{X}$, and prediction $\hat{\mathbf{y}}$ form a Markov chain $\mathbf{X} \rightarrow \mathbf{X} \rightarrow \hat{\mathbf{y}}$ since $P(\hat{\mathbf{y}}|\mathbf{s}, \mathbf{X}) = P(\hat{\mathbf{y}}|\mathbf{X})$ for the model with vanilla training. Based on data processing inequality, it is easy to obtain

$$
\begin{aligned}
I(\mathbf{s}; \hat{\mathbf{y}}_{MLP}) &= I(\mathbf{s}; \mathbf{X}) - I(\mathbf{s}; \mathbf{X}|\hat{\mathbf{y}}_{MLP}), \\
I(\mathbf{s}; \hat{\mathbf{y}}_{GCN}) &= I(\mathbf{s}; \tilde{\mathbf{X}}) - I(\mathbf{s}; \tilde{\mathbf{X}}|\hat{\mathbf{y}}_{GCN}).
\end{aligned}
\tag{16}
$$

In other words, when $I(\mathbf{s}; \mathbf{X}|\hat{\mathbf{y}}_{MLP}) = I(\mathbf{s}; \tilde{\mathbf{X}}|\hat{\mathbf{y}}_{GCN})$, the higher input data bias will lead to higher prediction bias. For one-layer MLP and one-layer GCN, if the condition in Theorem 2 are satisfied, the prediction bias of one-layer GCN is also larger than that of MLP.

## C.3 COMPARISON WITH CONCENTRATION PROPERTY IN GNN AND PERSISTENT HOMOLOGY

GNN's concentration property represents all node presentation convergence after stacking of aggregations (Nt & Maehara, 2019; Ma et al., 2022; Baranwal et al., 2021). There are differences between bias enhancement and concentration property in GNN:

- Definition. Concentration property means that the node representation of all nodes convergence after GNN aggregation. Bias enhancement represents the node representation for different sensitive groups that are more distinguished. In fact, such two properties are somehow contrary since perfect concentration leads to zero bias.

- Aggregation. For concentration property, only the normal features and topology are involved in the analysis. The high-level interpretation of concentration is that aggregation acts like a low-frequency filter and such an "average" effect leads to node representation convergence. In the sensitive homophily coefficient, we would like to clarify that the sensitive attributes are not included for node feature aggregation due to law restrictions. Even though the sensitive attribute is included in GNN aggregation, and all node representations are the same, it does not represent the bias enhancement (actually zero bias.). The bias in GNN represents the highly different node representations or predictions among different sensitive groups (defined by sensitive attributes).

- Comparison. We also provide the connection between concentration property and bias enhancement in GNN. The intuition of why GNN enhances the bias, high sensitive homophily represents that the nodes with the same sensitive attributes are connected with high probability. Considering concentration property, the node representation for the same sensitive attribute is more similar after aggregation, therefore leading to highly different representations for different sensitive attribute groups. Notice that such behavior only happens for high sensitive homophily coefficients and shallow GNN. When the node with different sensitive attribute groups is connected randomly, the bias enhancement would not happen or be insignificant due to random concentration. When adopting deep GNNs, all node representations converge and have no bias enhancement. Unfortunately, due to concentration property, shallow GNNs are mainly used in practice. As for high sensitive homophily coefficient, such a condition is usually satisfied in practice due to natural graph data property.

Additionally, there are several differences between our proposed optimization scheme and work (Carriere et al., 2021), including definition, dependence, and optimization:

- Definition: Persistent homology is a method for calculating the importance of topological features in the simplicial complex. For example, giving a set of points in a point cloud corresponding to a chair, the task is to detect the object from the points. In this case, there is no connection between any pair of points. Persistent homology is a tool to identify the topological feature (or connection patterns) via gradually building up the connection between points. However, for graphs, they are 1-simplex with explicit connection patterns defined by the set of edges, thus many properties from persistent homology will degenerate to the field of graph theory. For example, applying the persistent homology on the graph is equivalent to building maximum spanning trees (MSTs) using Kruskal algorithm (Kleinberg & Tardos, 2006), which is irrelevant to our proposed optimization scheme.
- Dependence. Persistent homology is generally related to sample features, as shown in the example Point cloud optimization of (Carriere et al., 2021). In other words, persistent homology somehow represents the topological features of all samples. Differentially, in the graph data we focused on, there are node normal attributes, sensitive attributes, and adjacency matrix (graph topology). Based on the definition, the sensitive homophily coefficient is related to sensitive attributes and the adjacency matrix. However, the optimized adjacency matrix is generally dependent on sensitive attributes.
- Optimization. The main challenge for persistent homology-based optimization is generally undifferentiable except in some special cases. (Carriere et al., 2021) develops a general framework to study the differentiability of the persistence of parametrized families of filtrations. In this way, under mild assumptions, stochastic subgradient descent algorithms can be applied to such functions to converge almost surely to a critical point. For our problem (2), the gradient of loss over topology is differentiable in general. The challenge falls in the constraint of an element value. In our solution, we use a gradient-based method to update the adjacency matrix via variables relaxation and then adopt project operation to satisfy such constraint.

## D  RELATED WORKS

**Graph neural networks.**  GNNs achieve state-of-the-art performance for various real-world applications. There are two categories in GNNs model backbones, including spectral-based and spatial-based GNNs. Spectral-based GNNs provide graph convolution operation together with feature transformation (Bruna et al., 2013; Defferrard et al., 2016; Henaff et al., 2015). Many spatial-based GNNs are also proposed to aggregate the neighbors' information, including graph attention network (GAT) (Veličković et al., 2018), GraphSAGE (Hamilton et al., 2017), SGC (Wu et al., 2019), APPNP (Klicpera et al., 2019), et al (Gao et al., 2018; Monti et al., 2017).

**Fairness-aware learning on graphs.**  Fairness in machine learning has attracted many research efforts to mitigate prediction bias (Chuang & Mroueh, 2020; Zhang et al., 2018; Du et al., 2021; Yurochkin & Sun, 2020; Jiang et al., 2022; Creager et al., 2019; Feldman et al., 2015). Fair walk (Rahman et al., 2019) is a fair version of random walk to learn fair node representation via revising neighborhood sampling. From the bias mitigation perspective, adversarial debiasing and contrastive learning are also developed for graph data. For example, works (Dai & Wang, 2021; Bose & Hamilton, 2019; Fisher et al., 2020) also adopt the adversary to predict the sensitive attribute given the node representation. fairness-aware representation learning is also developed via node

feature masking, graph topology rewires (Agarwal et al., 2021; Köse & Shen, 2021) for node classification or link prediction tasks (Laclau et al., 2021; Li et al., 2021). However, the inherent reason behind the observation that GNNs show higher prediction bias than MLP is still missing. In this work, we theoretically and experimentally reveal that many GNNs aggregation schemes boost node representation bias under topology bias. Furthermore, we develop a simple yet effective graph refinement method, named FairGR, to achieve fair prediction.

## E    DATASET STATISTICS

The data statistical information on three real-world datasets, including Pokec-n, Pokec-z, and NBA, are provided in Table 2. It is seen that the sensitive homophily is even higher than label homophily coefficient among three real-world datasets, which validates that real-world datasets are usually with large topology bias.

Table 2: Statistical Information on Datasets

| Dataset | # Nodes | # Node Features | # Edges | # Training Labels | # Training Sens | Label Homop | Sens Homop |
|---|---|---|---|---|---|---|---|
| Pokec-n | 66569 | 265 | 1034094 | 4398 | 500 | 73.23% | 95.30% |
| Pokec-z | 67796 | 276 | 1235916 | 5131 | 500 | 71.16% | 95.06% |
| NBA | 403 | 95 | 21242 | 156 | 246 | 39.22% | 72.37% |

## F    TRAINING ALGORITHMS

We summarize the training algorithm for FairGR and provide the pseudo codes in Algorithm 1.

---
**Algorithm 1:** FairGR Algorithm

---
**Input**    : Graph topology $\mathbf{A}$, label $\mathbf{y}$, and sensitive attribute $\mathbf{s}$; The total epochs $T$;
            Hyperparameters $\alpha$ and $\beta$.
**Output** : The fair graph topology $\tilde{\mathbf{A}}$.
1 **for** *epoch from* 1 *to* $T$ **do**
2    Calculate the gradient $\frac{\partial \mathcal{L}(\hat{\mathbf{A}}|\mathbf{s},\mathbf{y},\mathbf{A})}{\partial \tilde{\mathbf{A}}}$ ;
3    Update graph topology $\tilde{\mathbf{A}}$ using gradient descent;
4    Project graph topology $\tilde{\mathbf{A}}$ into a feasible region;
5 **end**

---

## G    MORE EXPERIMENTAL RESULTS

### G.1    MORE SYNTHETIC EXPERIMENTAL RESULTS

In this subsection, we provide more experimental results on **different** covariance matrix. Although our theory is only derived for the same covariance matrix, we still observe similar results for the case of **different** covariance matrix. For **node attribute generation**, we generate node attribute with Gaussian distribution $\mathcal{N}(\mu_1, \boldsymbol{\Sigma})$ and $\mathcal{N}(\mu_2, \boldsymbol{\Sigma})$ for node with binary sensitive attribute, respectively, where $\mu_1 = [0,1]$, $\mu_1 = [1,0]$ and $\boldsymbol{\Sigma} = \begin{bmatrix} 1 & 0 \\ 0 & 2 \end{bmatrix}$. We adopt the same evaluation metric and adjacency matrix generation scheme in Section 6.1

Figure 4 shows the DP difference during message passing with respect to sensitive homophily coefficient for different initial covariance matrices. We observe that a higher sensitive homophily coefficient generally leads to larger bias enhancement. Additionally, higher graph connection density $\rho_d$, larger node number $n$, and balanced sensitive attribute ratio $c$ correspond to higher bias enhancement, which is consistent with our theoretical analysis in Theorem 2.

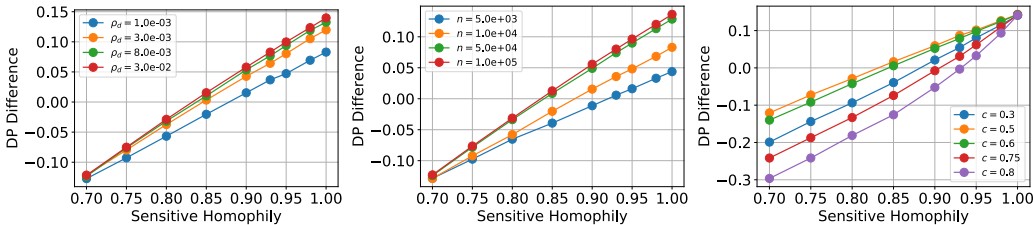

Figure 4: The difference of demographic parity for message passing with different initial covariance matrices. **Left:** DP difference for different graph connection density $\rho_d$ with sensitive attribute ratio $c = 0.5$ and number of nodes $n = 10^4$; **Middle:** DP difference for different number of nodes $n$ with sensitive attribute ratio $c = 0.5$ and graph connection density $\rho_d = 10^{-3}$; **Right:** DP difference for different sensitive attribute ratio $c$ with graph connection density $\rho_d = 10^{-3}$ and number of nodes $n = 10^4$.

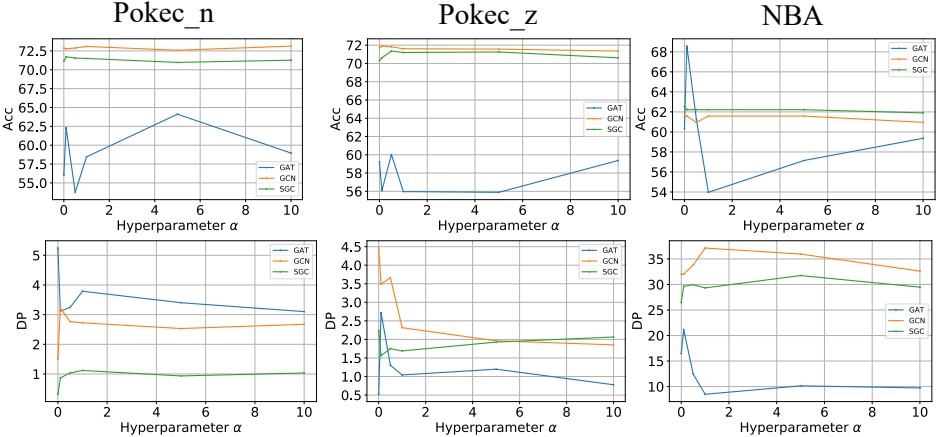

Figure 5: Ablation study on hyperparameter $\alpha$ in terms of DP and Acc across different GNN backbones on three real-world datasets.

## G.2 ABLATION STUDY FOR FAIRGR

For investigating the effect of hyperparameters $\alpha$ and $\beta$, we conduct experiments with different hyperparameters chosen from $\alpha = \{0.0, 0.1, 0.5, 1.0, 5.0, 10.0\}$ and $\beta = \{0.0, 0.1, 0.5, 1.0, 5.0, 10.0\}$ while the other one is selected as default. The default value for hyperparameters $\alpha$ and $\beta$ are $0.1$ and $0.5$, respectively. The results of hyperparamter study with respect to $\alpha$ and $\beta$ are shown in Figures 5 and 6, respectively. From these results, we can obtain the following observations:

- Hyprparameter $\alpha$ and $\beta$ demonstrate different influences on GNN backbone. For example, for Pokec-n dataset, hyperparameter $\alpha$ only shows a negligible influence on the accuracy of GCN and GAT, while significant for GAT. As for DP, the bias metric is more sensitive to $\alpha$ compared with accuracy.

- Hyprparameter $\alpha$ and $\beta$ demonstrate different influences on graph dataset. For example, GAT achieves the best accuracy and lowest DP with $\alpha = 0.1$ in NBA dataset, while achieving the lowest accuracy and highest DP in Pokec-n dataset. Such observation indicates the importance of hyperparameter tuning for different datasets.

## G.3 VANILLA TRAINING BEHAVIORS FOR GNNS AND MLP

For vanilla training across different GNN backbones, we plot the training curve with respect to epochs to investigate the training behaviors for GNNs and MLP. For the training behavior, GNNs,

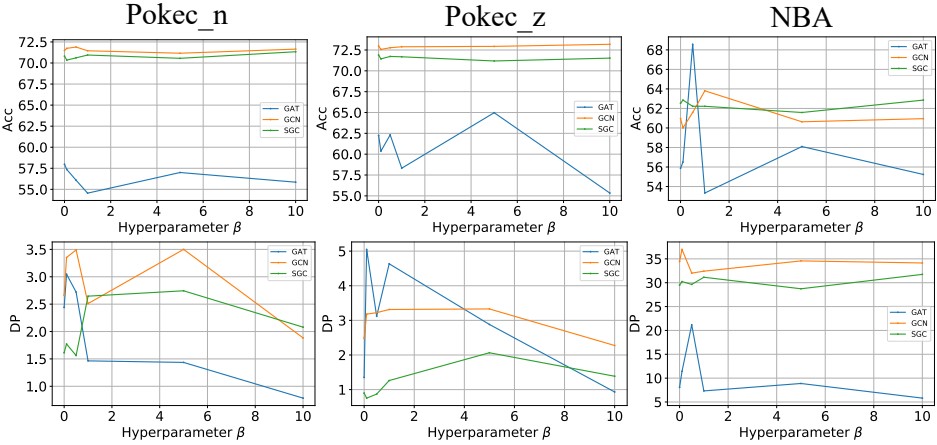

Figure 6: Ablation study on hyperparameter $\beta$ in terms of DP and Acc across different GNN backbones on three real-world datasets.

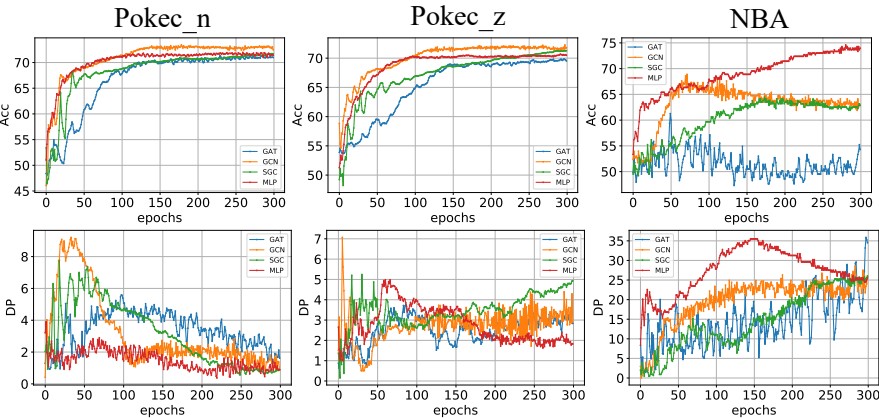

Figure 7: Accuracy (top) and DP (bottom) training curve w.r.t. epochs for different backbones, including GAT, GCN, SGC, and MLP, on three real-world datasets.

and MLP models both converge in terms of accuracy for the high label homophily dataset Pokec-n and Pokec-z dataset. For high sensitive homophily Pokec-n and Pokec-z dataset, GNNs demonstrate higher prediction than that MLP, while the prediction bias difference is relatively small for the low-sensitive-homophily dataset NBA.

### G.4 FAIRGR RESULTS ON VANILLA TRAINING

For vanilla training, Figure 8 shows the test accuracy and demographic parity curve during training for different GNNs backbones and whether FairGR is adopted for graph topology rewire. From these results, we can obtain the following observations:

- Different GNNs demonstrate different accuracy and demographic parity performance. For example, for Pokec-n dataset, GCN has the highest accuracy performance and lowest demographic parity, which implies that message passing also matters even for the same graph topology.

- Our proposed FairGR consistently achieves lower demographic parity and comparable accuracy performance on all datasets and backbones.

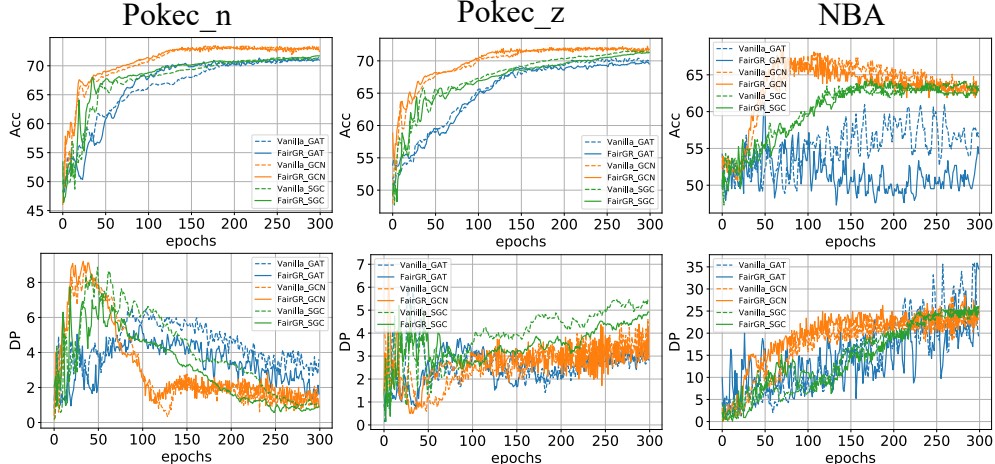

Figure 8: Accuracy (top) and DP (bottom) training curve w.r.t. epochs for different backbones, including GAT, GCN, SGC, and MLP, on three real-world datasets.

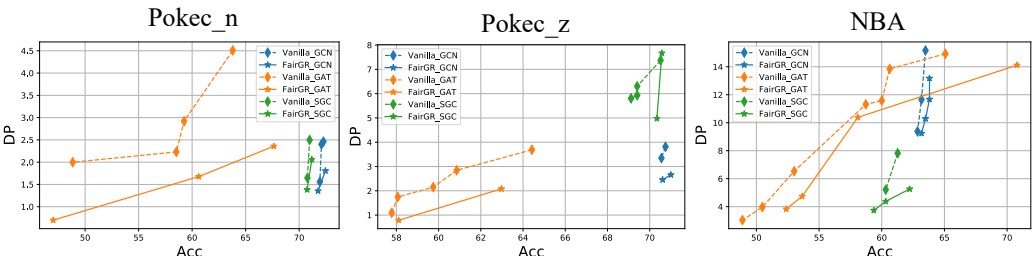

Figure 9: DP and Accuracy trade-off performance on three real-world datasets, including Poken-n, Pokec-z, and NBA, in (manifold) Fair Mixup setting.

### G.5 TRADEOFF PERFORMANCE ON FAIR MIXUP

We also demonstrate that FairGR can achieve better tradeoff performance for different GNN backbones with Fair mixup (Chuang & Mroueh, 2021) in Figure 9. Specifically, since input fair mixup requires calculating model prediction for mixed input batch, it is non-trivial to adopt input fair mixup in our experiments on the node classification task. This is because, for GNN aggregations of neighborhoods' information, the neighborhood information for the mixed input batch is missing. Instead, we adopt manifold fair mixup for the logit layer in our experiments. Experimental results show that FairGR can achieve better accuracy-fairness tradeoff performance for many GNNs backbones on three datasets.

## H FUTURE WORK

There are two lines of follow-up research directions. Firstly, The generalization of the theoretical analysis on why aggregation enhances bias in GNN can be further extended. As a pilot study, we theoretically investigate why this phenomenon happens for GCN-like aggregation under random graph topology generated by stochastic block model and Gaussian mixture feature distribution. The more general analysis of other aggregation operations, random graph models, and other feature distributions can be extended. The other line focuses on the graph topology rewire algorithmic perspective, including improving the efficiency of FairGR and effectiveness via designing different objectives for graph topology rewire.

