# OpenReview forum: "Topology Matters in Fair Graph Learning: a Theoretical Pilot Study"
_ICLR.cc/2023/Conference — Submitted to ICLR 2023_

### Official Review · Reviewer_bTV8 · 2022-10-24

**Confidence:** 3
**Correctness:** 3
**Technical Novelty And Significance:** 3
**Empirical Novelty And Significance:** 3
**Recommendation:** 6

**Clarity, Quality, Novelty And Reproducibility:**

The proposed method is simple, but the fact that it uses an analysis of why Bias increases as a basis for its proposal is an originality novelty. However, the quality is poor and unclear at the stage of showing the effect of the proposed method.

**Strength And Weaknesses:**

Weakness
- The biggest problem is that the content is not complete, especially after Section.4. This makes the evaluation of the proposed methodology difficult. If we try to interpret and understand what is written in a favorable manner, we can assume that the claims are probably correct, but we cannot accept them as being of publishable quality. I think the assignment settings and methods are interesting and should be evaluated carefully after they are completed.
- Specifically, the major problems are as follows
	- I'm not sure what equation (2) means, but since it seems to be an optimization problem, perhaps
$$\min L s.t.\tilde{A}={ij}\in\{0,1\}$$
where $L=H1+\alpha H2+\beta H3$?
	- Regarding Table 1, it says "To validate the effect of bias enhancement of GNNs," but it is not a comparison with the proposed method. The intent may be to confirm the current status, but in that case the text is confusing, and furthermore, there should be a separate comparison with the proposed method.
	- Figure. 6 is reffered, but it is not appropriate to refer the figure in the suppliment. Figure.3 may be correct instead of Figure.6. It seems that a comparison with the proposed method may be more appropriate for the content in one piece. Therefore, it may be appropriate and intended to publish Fig. 6 in the main body. However, even in that Fig. 6, two of the three are compared to the proposed method, but the quality of the remaining one is poor, with no comparison.
	- It may not be required, but it is not reader friendly to have RELATED WORK at the end.

**Summary Of The Paper:**

This paper proposes a way to solve the problem that GNNs tend to increase Bias. First, the authors give a mathematical interpretation of why GNNs increase Bias. Based on the causes obtained from this interpretation, we propose a method to suppress the increase in bias that occurs during message passing. Experimental results showing the effectiveness of the method are also presented.

**Summary Of The Review:**

Issues related to Bias are important issues in AI operations, and the fact that GNNs tend to increase Bias is problematic, and it is a very important task to analyze these factors and then present improvement methods. The analysis of the causes of GNNs increasing Bias seems justified, and the improvement methods are highly convincing.

On the other hand, since the quality of the structure since the proposal of the methodology is in a low state and the information to evaluate the methodology is unclear, the text should first be refined and the evaluation of the methodology should be clarified. We believe this will result in something that can be made public.

**Conclusion following discussion with the authors**

The problems with the sentence structure that were hindering my understanding have been largely remedied. As noted before, the value of the methodology itself is positive. For this reason, I am raising the score.

---

> ### Author Response · Authors · 2022-11-16
> **Response to Reviewer bTV8**
>
> We thank the reviewer for the constructive comments and appreciate the reviewer for recognizing the novelty of our work.
>
> Q1: The content is not complete, especially after Section.4. This makes the evaluation of the proposed methodology difficult.
>
> A1: Thanks for pointing out the incomplete content on evaluation. FairGR is a ** data preprocessing** method and aims to **rewire graph topology** for fair prediction. For the evaluation part, we compared the prediction performance (including accuracy and fairness) using original graph topology and rewired graph topology (FairGR) across multiple GNN backbones. In this way, we can validate whether FairGR can achieve fair prediction via rewiring graph topology. Please see Section 4 for more details.
>
> Q2: I'm not sure what equation (2) means, but since it seems to be an optimization problem.
>
> A2: Equation (2) is an optimization problem. As the reviewer suggested, to avoid confusion, we first define the loss function and then introduce the optimization problem.
>
> Q3: Regarding Table 1, it says "To validate the effect of bias enhancement of GNNs," but it is not a comparison with the proposed method. The intent may be to confirm the current status, but in that case, the text is confusing, and there should be a separate comparison with the proposed method.
>
> A3: As for FairGR, we add experimental results on FairGR coupled with different GNN backbones in Table 1.
>
> Q4: Figure. 6 is referred to, but it is not appropriate to refer to the figure in the supplement. Even in that Fig. 6, two of the three are compared to the proposed method, but the quality of the remaining one is poor, with no comparison. Overall,  the quality of the structure and the proposal of the methodology is in a low state and the information to evaluate the methodology is unclear.
>
> A4: We move Figures. 3 and 6 to  Appendix G.3 and G.4 due to limited space. Specifically, we compare the vanilla training behavior with or without graph topology rewire. As for trade-off performance comparison, we encourage the reviewer to focus on Figures. 4 and 7. This is because most of the fairness methods can mitigate bias while inevitably degrading accuracy performance. Therefore, it is generally **hard to comprehensively compare algorithms** for such incomparable cases. In this paper, we also provide **Pareto frontier** to comprehensively compared the tradeoff performance via adjusting different hyperparameters. For example, FairGR (a preprocessing method) is **compatible with methods based on fair loss**, including adding regularization, adversarial debiasing, and Fair Mixup. We demonstrate that our methods can be compatible with these methods and achieve better tradeoff performance.

---

> > ### Comment · Reviewer_bTV8 · 2022-11-18
> > **Thank you for the revision**
> >
> > Hi Authors,
> >
> > Thank you for the clarification of my concerns and the revision of the paper. I now have clarity on an issue that was hindering my understanding of the content. I  will now review the revised paper again and modify the evaluation appropriately based on the value of the results.
> >
> > Best

---

> > > ### Author Response · Authors · 2022-11-27
> > > **Thanks for the re-assessment with positive feedback!!**
> > >
> > > Dear Reviewer bTV8:
> > >
> > > Thanks for your positive re-assessment of our work. We enjoyed the fruitful discussion with you. Please do let us know if there is anything that you believe we can do to improve this work!
> > >
> > > Best, Authors

---

### Official Review · Reviewer_K1rB · 2022-10-24

**Confidence:** 2
**Correctness:** 3
**Technical Novelty And Significance:** 2
**Empirical Novelty And Significance:** 1
**Recommendation:** 3

**Clarity, Quality, Novelty And Reproducibility:**

I think the paper can be greatly improved on clarity. The main analysis is also similar with existing work.

**Strength And Weaknesses:**

Strength:
1. The analysis is theoretically sound with appropriate discussion and remarks.
2. The proposed FairGR coupled with proposed representation bias closely.

Weakness:
1. The motivation of FairGR is unclear. From Table 1, we observe GNN is just on par or slight better than MLP while the fairness measure is a lot worse. Does that mean Fair learning on GNN is not necessary on these datasets ? Furthermore, whether the proposed FairGR can reduce $\Delta_\text{DP}$ and $\Delta_\text{EO}$ to the level of MLP in Table 1.
2. A lot of notations in the paper are not clear or assuming readers with relevant knowledge. For example, in definition 2, "random graph sampled from ()", after reading the whole paper, I realize the generalization process is exact the contextual stochastic block model (CSBM), it is very confusing in this definition about how A and X are generated.
3. The novelty is limited. Based on my understanding, if we treat sensitive feature as the only node feature, the paper's contribution is not that different from [1][2], which both talks about GNN's concentration property.
[1] IS HOMOPHILY A NECESSITY FOR GRAPH NEURAL NETWORKS? ICLR 22'
[2] Graph Convolution for Semi-Supervised Classification: Improved Linear Separability and Out-of-Distribution Generalization ICML 21'

**Summary Of The Paper:**

This paper studies prediction bias amplification of graph neural networks using contextualized stochastic block model. Specifically, their analysis utilizes a distribution distance as bias measurement and discusses when $\Delta$ bias is enhanced regarding homophily edge ratio, graph size and density. Based on it, the author proposes a regularization framework FairGR to rewire the graph structure and reduce the representation bias. Experiment shows FairGR achieves both good accuracy and low bias.

**Summary Of The Review:**

Overall, the idea of the paper is clear and proposed method is straightforward. My main concern is around the novelty and experiments. First of all, the representation bias is not that different with other homophily understanding on GNNs. Second, the selected datasets cannot convince that proposed method is significant for fairness learning: (1) GNN is not better than MLP on accuracy (2) I suspect the bias of FairGR is still larger than MLP.

---

> ### Author Response · Authors · 2022-11-16
> **Response to Review K1rB [Part 1/2: Experimental results in Table 1]**
>
> We thank the reviewer for the constructive comments. It seems that you have a negative attitude toward our work. To address your concerns, we give the following point-to-point responses. We believe that if you read it carefully, you will change your attitude.
>
> Q1: The **motivation** of FairGR is unclear. From Table 1, we observe GNN is just on par or slightly better than MLP while the fairness measure is a lot worse. Does that mean Fair learning on GNN is not necessary on these datasets? Furthermore, whether the proposed FairGR can reduce $\Delta_{DP}$ and $\Delta_{EO}$ to the level of MLP in Table 1.
>
> A1: We would like to clarify that the results in Table 1 only show the results of vanilla GNNs and MLPs, which demonstrates current GNNs using the original graph topology will amplify prediction bias. These results demonstrate the fairness issue of many representative GNNs. As for the marginal improvement of GNN, I would like to clarify that such improvement **depends on the dataset and GNNs backbone**. For example, several GNNs, such as GCN and SGC, can even be **worse** than MLP in **low-label homophily dataset**. In this work, we mainly focus on topology perspective and propose a **graph topology rewire** method, which is **orthogonal to GNN backbones design**.
> We add more experimental results of FairGR with different GNN backbones in Table 1. As for MLP, FairGR can not improve MLP model performance since FairGR only rewires graph topology and the graph topology is not adopted in MLP. Additionally, the proposed method is also **orthogonal to fair learning methods** in GNN, such as adding regularization, adversarial debiasing, and Fair Mixup. FairGR can even further improve tradeoff performance across these fair learning methods.
>
>
> Q2: A lot of notations in the paper are not clear or assume readers with relevant knowledge. For example, in definition 2, "random graph sampled from ()", after reading the whole paper, I realize the generalization process is exactly the contextual stochastic block model (CSBM), it is very confusing in this definition about how $\textbf{A}$ and $\textbf{X}$ are generated.
>
> A2: We have gone through the manuscript and revised the notations to make the content clear.

---

> ### Author Response · Authors · 2022-11-16
> **Response to Review K1rB [Part 2/2: Novelty and Experimental results]**
>
> Q3: The **novelty is limited**. Based on my understanding, if we treat sensitive feature as the only node feature, the paper's contribution is not that different from [1][2], which both talks about GNN's concentration property.
>
> [1] IS HOMOPHILY A NECESSITY FOR GRAPH NEURAL NETWORKS? ICLR 22'
>
> [2] Graph Convolution for Semi-Supervised Classification: Improved Linear Separability and Out-of-Distribution Generalization ICML 21'.
>
> A3:We would like to clarify that our contributions are sufficiently novel. Specifically, we conduct a **theoretical pilot study** on fairness in graph on **wh**y aggregation operation in representative GNNs accumulates bias in node representation due to topology bias. As for GNN's **concentration** property, we claim there are differences between bias enhancement and concentration property in GNN.
> * **Definition**. Concentration property means that the **node representation** of all nodes **convergence** after GNN aggregation. Bias enhancement represents the node representation for different sensitive groups that are **more distinguished**. Such two properties are somehow **contrary since perfect concentration leads to zero bias**.
> * **Aggregation**. For concentration property, only the **normal features and topology** are involved in the analysis. The high-level interpretation of concentration is that aggregation acts like a **low-frequency filter** and such an "average" effect leads to node representation convergence. In sensitive homophily coefficient, we would like to clarify that the sensitive attributes are **not included** for node feature aggregation due to **law restriction**. Even though the sensitive attribute is included in GNN aggregation, and all node representations are the same, it **does not** represent the bias enhancement (actually zero bias). The bias in GNN represents the highly different node representations or predictions among different sensitive groups (defined by sensitive attributes).
> * **Comparison**. We also provide the **connection** between concentration property and bias enhancement in GNN. As we mentioned in Introduction Section on the intuition of why GNN enhances the bias, high sensitive homophily represents that the nodes with the same sensitive attributes are connected with high probability. Considering concentration property, the node representation for the **same** sensitive attribute is **more similar** after aggregation and therefore leading to highly different representations for different sensitive attribute groups. Notice that such behavior **only happens** for **high sensitive homophily coefficients and shallow GNN**. When the node with different sensitive attributes groups are connected randomly, the bias enhancement would not happen or be insignificant due to random concentration. When adopting deep GNNs, all node representations converge, and no bias enhancement. Unfortunately, due to concentration property, shallow GNNs are mainly used in practice. As for high sensitive homophily coefficient, such a condition is usually satisfied in practice due to natural graph data property.
> Please see Appendix C.3 for more details.
>
> Q4: The selected datasets cannot convince that the proposed method is significant for fairness learning: (1) GNN is not better than MLP on accuracy (2) I suspect the bias of FairGR is still larger than MLP.
>
> A4: The selected datasets are usually adopted for fairness in graph [C,D,E] and there are only **limited graph datasets** with sensitive attributes. As for that GNN is not better than MLP in accuracy, it is **not rare** in current literature and it depends on GNN backbone and graph data properties such as label homophily coefficient. As for FairGR, we add experimental results on FairGR coupled with different GNN backbones in Table 1.
>
> [C] Enyan Dai and Suhang Wang. Say no to the discrimination: Learning fair graph neural networks with
> limited sensitive attribute information. In Proceedings of the 14th ACM International Conference
> on Web Search and Data Mining, pp. 680–688, 2021.
>
> [D]Ö. D. Köse and Y. Shen*, "Fairness-aware Graph Contrastive Learning," IEEE Transactions on Signal and Information Processing over Networks, May 2022.
>
> [E]Zhimeng Jiang, Xiaotian Han, Chao Fan, Fan Yang, Ali Mostafavi, and Xia Hu. Generalized
> demographic parity for group fairness. In International Conference on Learning Representations,
> 2022.

---

> > ### Comment · Reviewer_K1rB · 2022-11-18
> > **Thanks author for the clarifications of my questions.**
> >
> > I have read through the rebuttal. Some of my questions are address such as Q1 and part of Q3. However I still hold my reservations on recommending this work. First of all, in updated Table 1, the author provide both accuracy measures and fairness measure of the proposed method "*-GR". We can see that MLP is on par or better than before/after rewired graph, such that this experiment cannot stress the significance of the problem. I do not agree with the logic that there are only "limited graph datasets with sensitive attributes" so we use the dataset in the paper. Based on my understanding, if the sensitive issue is not severe than the motivation of the problem could be weaken as well.

---

> > > ### Author Response · Authors · 2022-11-18
> > > **Response to Reviewer K1rB [Part 1/2: The experiments on the accuracy and fairness metrics]**
> > >
> > > We thank the reviewer for the timely comments.
> > >
> > > Q1: In updated Table 1, the author provides both accuracy measures and fairness measures of the proposed method "*-GR". We can see that MLP is on par or better than before/after rewired graph, such that this experiment cannot stress the significance of the problem.
> > >
> > > A1: In this work, we aim to validate the idea that topology in graph data is important in fair graph learning. In table 1, "*-GR" method can achieve less $\Delta_{DP}$ and $\Delta_{EO}$ than the vanilla method, which has already validated the idea of topology matters. For fairness research, most fairness learning methods cannot achieve better accuracy and lower bias than the vanilla method. Instead, the fairness method mainly focused on mitigating bias with comparable or slightly worse accuracy performance.
> > >
> > > As for the comparison with MLP, we agree that MLP can achieve low bias. We would like to argue that this is because there is no topology information adopted in the prediction, which also validates the idea of topology matters in fair graph learning. Additionally, GCN-GR achieves better accuracy, lower (slightly higher) $\Delta_{DP}$, and lower $\Delta_{EO}$ in Pokec-n (Pokec-z) dataset. In NBA dataset, GCN-GR, GAT-GR, and SGC-GR methods achieve lower $\Delta_{DP}$ and $\Delta_{EO}$ with lower accuracy. We would like to remind the reviewer that it is unrealistic to expect a new method can achieve better accuracy and lower bias simultaneously since there is usually **no free lunch**. To comprehensively evaluate the accuracy and bias tradeoff performance, we provide the Pareto frontier performance in Figures 3 and 9. The experimental results show that GNNs with graph rewire can achieve better tradeoff performance than their counterparts.
> > >
> > > Q2:I do not agree with the logic that there are only "limited graph datasets with sensitive attributes" so we use the dataset in the paper. Based on my understanding, if the sensitive issue is not severe then the motivation of the problem could be weakened as well.
> > >
> > >
> > > A2: As we mentioned in the previous response, we follow previous works [A, B, C] to conduct experiments in Pokec-z, Pokec-n, and NBA datasets. To the best of our knowledge, these datasets are the only available public graph datasets with sensitive attributes. Many datasets in graph learning, such as Cora and OGB datasets, are infeasible in fair graph learning since the sensitive attribute is not available in these datasets while is necessary to evaluate prediction bias.
> > >
> > > As for the motivation, we argue that it is **well-known and empirically observed** that GNNs achieve higher bias than MLP. However, it is still unclear why this phenomenon happens. In this work, we theoretically investigate why this phenomenon happens as a pilot study. We believe that the motivation for this work is clear.
> > >
> > >
> > > [A] Enyan Dai and Suhang Wang. Say no to the discrimination: Learning fair graph neural networks with limited sensitive attribute information. In Proceedings of the 14th ACM International Conference on Web Search and Data Mining, pp. 680–688, 2021.
> > >
> > > [B]Ö. D. Köse and Y. Shen*, "Fairness-aware Graph Contrastive Learning," IEEE Transactions on Signal and Information Processing over Networks, May 2022.
> > >
> > > [C]Zhimeng Jiang, Xiaotian Han, Chao Fan, Fan Yang, Ali Mostafavi, and Xia Hu. Generalized demographic parity for group fairness. In International Conference on Learning Representations, 2022.

---

> > > ### Author Response · Authors · 2022-11-21
> > > **Response to Reviewer K1rB [Part 2/2: Misunderstandings on the limited graph datasets and motivations]**
> > >
> > > Thanks to the reviewer for the constructive comments to improve the quality of this work. We would like to make more justifications for the concern of "limited amount of graphs with sensitive attributes" since this comment mislead another reviewer. We believe that there are **misunderstandings** for this concern, especially regarding the graph dataset and the motivation of our work.
> > >
> > > * **Graph datasets with sensitive attributes**: we agree that the number of **public** graph datasets with sensitive attributes (available sensitive attributes for all nodes) is limited in current fair graph learning research. The main reason is that the sensitive attribute of each node (person in the social network) is hard to collect due to privacy constraints. Additionally, we **mainly focus** on the scenario in that the sensitive attribute of **all** nodes is available, which makes such datasets harder to collect. The more practical situation is that only the sensitive attribute of a part of nodes is available. We would like to clarify that our main contribution is to theoretically investigate why GNN achieves higher bias than MLP as a pilot study, and we leave the **extension to limited available sensitive attributes for future work**. Meanwhile, there are many other public tabular data with sensitive attributes, such as Adult, German, and Credit. However, there is no topology information in these datasets. In this work, we follow previous work [A, B, C] to conduct experiments in fair graph learning. To the best of our knowledge, these datasets are the only available public graph datasets with sensitive attributes. The limited public graph datasets exist for current fair graph learning research, which indicates that more public graph data with sensitive attributes would be valuable. However, there is **no relationship between this concern and the motivation of our work**.
> > > * **Motivation**: The phenomenon that GNNs achieve higher bias than MLP is well-known and empirically observed in current fair graph learning literature. However, most previous works mainly focus on how to mitigate bias for graph learning. The **reason behind such a phenomenon** is still unclear. Motivated by such observation, as a pilot study, we theoretically investigate why such a phenomenon happens. Specifically, under contextualized stochastic block model, we characterize a sufficient condition on statistical information of graph data to guarantee the phenomenon happens. We believe that the motivation of this work is clear, and hope reviewers focus on our main contribution parts.
> > >
> > > In summary, we agree that there are limited graph datasets with sensitive attributes due to privacy constrain. Many graph datasets, such as Cora, Reddit, and OGB, can not be applied in fair graph learning research since there are no sensitive attributes of each node. However, such an issue is mainly related to the **entire** fair graph learning research and indicates that more graph datasets with sensitive attributes are valuable. In our work, we don't claim the contribution for providing dataset resources, and this part is **not related to the motivation** of our work. The main contribution is to, as a pilot study, theoretically investigate why such a phenomenon happens. We hope reviewers can focus on our main contribution parts.
> > >
> > > [A] Enyan Dai and Suhang Wang. Say no to the discrimination: Learning fair graph neural networks with limited sensitive attribute information. In Proceedings of the 14th ACM International Conference on Web Search and Data Mining, pp. 680–688, 2021.
> > >
> > > [B]Ö. D. Köse and Y. Shen*, "Fairness-aware Graph Contrastive Learning," IEEE Transactions on Signal and Information Processing over Networks, May 2022.
> > >
> > > [C]Zhimeng Jiang, Xiaotian Han, Chao Fan, Fan Yang, Ali Mostafavi, and Xia Hu. Generalized demographic parity for group fairness. In International Conference on Learning Representations, 2022.

---

> > > ### Author Response · Authors · 2022-11-29
> > > **Looking forward to discussing**
> > >
> > > Dear Reviewer K1rB,
> > >
> > > Thank you again for the constructive and valuable comments. As we are nearing the end of the discussion phase, we would like to know if our response has addressed your questions, especially regarding the motivations. If you have any further questions, we would be more than happy to address them.
> > >
> > > Thanks,
> > > Authors

---

> > > ### Author Response · Authors · 2022-12-11
> > > **To Reviewer K1rB: Are there more questions/concerns left?**
> > >
> > > Dear Reviewer K1rB,
> > >
> > > We appreciate your constructive and valuable comments to improve our work. We believe that the remaining concerns on the limited graph datasets and motivations are mainly from misunderstanding. We try to tackle these concerns in the rebuttal. As we are nearing the end of the discussion phase, we would like to know if our response has addressed your remaining concerns. It will be our pleasure to receive your follow-up feedback.
> > >
> > > Thanks, Authors

---

### Official Review · Reviewer_qbDf · 2022-10-24

**Confidence:** 2
**Correctness:** 4
**Technical Novelty And Significance:** 2
**Empirical Novelty And Significance:** 2
**Recommendation:** 6

**Clarity, Quality, Novelty And Reproducibility:**

I am not very knowledgeable in graph neural networks, but the proposed approach looks quite original and easy to reproduce to me.

**Strength And Weaknesses:**

Strengths:
---The paper is overall well written but a bit hard to follow
---To my knowledge, it addresses an interesting problem about graph neural nets
---The experiments are convincing

Weaknesses:
---Some parts of the method have a little bit of a heuristic feeling; for instance, I am wondering if other models could be used (instead of gaussian mixtures) for random graphs? Is there a reason (other than mathematical computations) for using such random models?
---Similarly, there is no discussion about the optimization properties of (2). Would it be possible to show some kind of convergence results, or to prove anything concerning the limit point?
---Graph topology is often characterized with persistent homology. Are there links between the proposed optimization scheme and recent papers about topology optimization based on persistence diagrams? (see for instance http://proceedings.mlr.press/v139/carriere21a.html) It would be nice to add some discussion on this point.


**Summary Of The Paper:**

In this article, the authors propose to study and improve fairness in graph neural networks. More precisely, they provide conditions under which the graph aggregation steps increase prediction biases, and they propose an optimization problem that allows to find an optimal graph topology that reduces biases while preserving topology information. They finally demonstrate the efficiency of their approach on several experiments.

**Summary Of The Review:**

Overall, the paper looks good to me, even though I am no expert in graph neural networks.

---

> ### Author Response · Authors · 2022-11-16
> **Response to Review qbDf [Part 1/2: Methodology discussion]**
>
> We thank the reviewer for the constructive comments and appreciate the reviewer for recognizing the novelty of our work.
>
> Q1: Some parts of the method have a little bit of a **heuristic** feeling. Why not use other models (instead of gaussian mixtures) for the random graph? Is there a reason (other than mathematical computations) for using such random models?
>
> A1: The theory of random graphs is **widely** used to model and analyze most complex networks (e.g., social networks, World Wide Web, and biological networks) for studying their behavior and for capturing the uncertainty and the lack of regularity [A]. We synthesize random graph as Definition 2 since the **graph statistical information,** including the number of nodes $n$, the edge density $\rho_d$, sensitive homophily coefficient $\epsilon_{sens}$, and sensitive
> group ratio $c$, are the most **fundamental and target** properties. The random graph synthesization in Definition 2 is the most **simple** way. We concede that our analysis is based on synthesized graph topology in Definition 2 and synthesized node features with Gaussian mixtures distribution. We would like to clarify that, as a **pilot study**, we choose the most common-used graph topology and node feature distributions in the analysis. For the **extension** to other random graph models or node feature distributions, we leave the analysis in the future work since it may require highly different analysis method. Please refer to Appendix H for more discussion on future work.
>
>
> [A] Van Der Hofstad, R. (2016). Random graphs and complex networks (Vol. 43). Cambridge university press.
>
> Q2: There is no discussion about the optimization properties of (2), such as convergence results.
>
> A2: We would like to clarify that problem (2) is a discrete optimization problem, and we solve this problem via the **relaxation** of variables and **projection operation** in PGD. Generally speaking, it is **intractable** to provide convergence results for a such large-scale optimization problem. We direct use PGD as a solver. Additionally, we would like to clarify our contribution mainly focus on the t**heoretical analysis between fairness/bias and graph topology**. Based on the theoretical insight, we develop a **simple yet effective** FairGR algorithm to obtain rewired graph topology. The convergence analysis or acceleration can be left for future work.

---

> ### Author Response · Authors · 2022-11-16
> **Response to Review qbDf [Part 2/2: Comparison with persistency homology]**
>
> Q3: Graph topology is often characterized by **persistent homology**. Are there links between the proposed optimization scheme and recent papers about topology optimization based on persistence diagrams?
>
> A3: Thanks to the reviewer for pointing out persistent homology. Although we are not experts in persistent homology, to the best of our knowledge, the difference between our proposed optimization scheme with [B] are three-fold:
>
> * **Definition**: Persistent homology is a method for calculating the **importance of topological features** in the simplicial complex. For example, giving a set of points in a point cloud corresponding to a chair, the task is to detect the object from the points. In this case, there is **no connection** between any pair of points. Persistent homology is a tool to identify the topological feature (or connection patterns) via **gradually** building up the connection between points. However, for graphs, they are **1-simplex** with **explicit connection patterns** defined by the set of edges, thus many properties from persistent homology will degenerate to the field of graph theory. For example, applying the persistent homology on the graph is equivalent to building maximum spanning trees (MSTs) using Kruskal algorithm [B], which is **irrelevant** to our proposed optimization scheme.
>
>
> * **Dependence**. Persistent homology is generally related to **sample features**, as shown in the example Point cloud optimization of [C]. In other words, persistent homology somehow represents the topological features of all samples. Differentially, in the graph data we focused on, there are node normal attributes, sensitive attributes, and adjacency matrix (graph topology). Based on the definition, the sensitive homophily coefficient is related to sensitive attributes and the adjacency matrix**. However, the optimized adjacency matrix is generally dependent on sensitive attributes.
>
> * **Optimization**. The main challenge for persistent homology-based optimization is generally **undifferentiable** except in some special cases. [C] develops a general framework to study the differentiability of the persistence of parametrized families of filtrations. In this way, under mild assumptions, stochastic subgradient descent algorithms can be applied to such functions to converge almost surely to a critical point. For our problem (2), the gradient of loss over topology is **differentiable** in general. The challenge falls in the **constraint of the binary element value**. In our solution, we use a gradient-based method to update the adjacency matrix via variables relaxation and then adopt project operation to satisfy such constraint.
>
> Please see more details in Appendix C.3.
>
> [B] Kleinberg, Jon, and Eva Tardos. Algorithm design. Pearson Education India, 2006.
>
> [C] Carriere, Mathieu, et al. "Optimizing persistent homology based functions." International conference on machine learning. PMLR, 2021.

---

> ### Author Response · Authors · 2022-11-29
> **Looking forward to your response**
>
> Dear Reviewer qbDf,
>
> Thank you again for the constructive and valuable comments. As we are nearing the end of the discussion phase, we would like to know if our response has addressed your questions. If you have any further questions, we would be more than happy to address them.
>
> Thanks, Authors

---

> ### Comment · Reviewer_qbDf · 2022-12-05
> **Response**
>
> Thank you for your answers and clarifications.
>
> A quick note: persistent homology amounts to maximum spanning tree only in homology dimension 0, if you want to compute (extended) persistent homology in dimension 1, you need tools that are not from graph theory.

---

> > ### Author Response · Authors · 2022-12-10
> > **Response to Review qbDf**
> >
> > We thank the reviewer for the constructive comments and appreciate the reviewer for recognizing the novelty of our work. In this paper, we consider graph data with explicit connection patterns (adjacency matrix), which is 1-simplex with explicit connection patterns (defined by the set of edges). Therefore, our contribution is not related to persistent homology. Instead,  we conduct the analysis (why GNNs aggregation leads to higher prediction bias) based on contextual stochastic block model (CSBM) and probability theory.

---

### Official Review · Reviewer_SmbS · 2022-10-25

**Confidence:** 3
**Correctness:** 4
**Technical Novelty And Significance:** 4
**Empirical Novelty And Significance:** 3
**Recommendation:** 6

**Clarity, Quality, Novelty And Reproducibility:**

Clarity: Most of the paper is clear except for some experimental setups that are a little vague.

Quality: The theoretical analysis is concrete but the writing can be further improved.

Novelty: The idea of theoretically analyzing the fairness-aware graph learning approaches has novelty and merits.

Reproducibility: The reproducibility is relatively weak with no codes provided.


**Strength And Weaknesses:**

Strengths:
* The investigation of fairness-aware message-passing from the topology view is interesting.
* The theoretical foundations on node representation bias amplification are concrete with insights. Experiments are carefully designed and are conducted on both synthetic and real-world graph datasets.
* The paper is generally well-written (though with typos and grammar errors) and almost clear everywhere.

Weaknesses:
* Some typos appear as I walk through the manuscript. For example, by that fact -> by the fact, the last section on page 1; many enhance -> may enhance, the last section on page 5. I suggest the authors do thorough proofreading during the rebuttal.

* The scalability of the proposed FairGR is not demonstrated and the optimization strategy is briefly described without computational analysis, which is important to employ in practical scenarios for graphs with massive nodes. Meanwhile, the sensitivity analysis of hyperparameters $\alpha$ and $\beta$ is suggested to be included for better demonstrating the contributions of label homophily coefficient and graph topology perturbation during learning.

* Synthetic experiments from Figure 2 are somehow vague and confusing. Since it shows the DP difference during message passing with respect to the sensitive homophily coefficient, what is the instantiation of GNNs used here for the message-passing process? Does MLP also show similar behavior regarding DP difference and sensitivity homophily here?


**Summary Of The Paper:**

This paper studies the problem of fairness-aware learning on graphs from the perspective of graph topology. According to the authors, this is the first work that theoretically understands the different roles graph statistical information played in the node representation bias amplification during message passing. Then a graph refinement method FairGR is proposed based on the theoretical results to reduce the sensitive homophily accordingly.
Experiments on both synthetic and real-world datasets show the effectiveness of FairGR by achieving better tradeoff performance over three representative baselines.


**Summary Of The Review:**

In summary, my concerns are mainly from two aspects:
* The scalability of FairGR could be further demonstrated through complexity analysis. The sensitivity of hyperparameters should be addressed.
* The details of experimental setups could be further explained for more convincing empirical support to the theoretical foundations.

---

> ### Author Response · Authors · 2022-11-16
> **Response to Review SmbS**
>
> We thank the reviewer for the constructive comments and appreciate the reviewer for recognition of the novelty of our work.
>
> Q1: The **scalability** of FairGR could be further demonstrated through complexity analysis. The **sensitivity of hyperparameters** should be addressed.
>
> A1: We add a paragraph on the computation complexity analysis for FairGR. Specifically, denote the number of training nodes and update iterations as $N$ and $T$, respectively. Then the computation complexity for gradient computation and projection of PGD are both $O(n_{train}^2)$. The total **computation complexity** to obtain the final rewired graph topology is given by $O(Tn_{train}^2)$. The **memory consumption** is $O(n_{train}^2)$ due to the storage of graph topology gradient. We clarify that our main contribution falls in the **theoretical pilot study on the relationship between fairness and graph topology**, and we leave the efficiency improvement in future work. As for the hyperparameter study of **$\alpha$ and $\beta$**, we add the experimental results in Appendix G.2.
>
> Q2: The **details of experimental setups**, especially for the synthetic experiments in Figure 2, could be further explained for more convincing empirical support to the theoretical foundations.
>
> A2: We are sorry for the misunderstanding on the experimental setup. For the synthetic experiments, we aim to validate the theoretical results that theoretically demonstrate why the GCN-like message passing **operation** in GNN enhances bias in Section 3. Therefore, in the synthetic experiments, we demonstrate the relation between DP difference across **GCN-like message passing operation** and sensitive homophily coefficient. Note that we only do a theoretical study in GCN-like message-passing operation as a pilot study. The investigation of other GNN aggregation operations (such as GraghSAGE-like operation) and GNN models may require different techniques and can be further conducted in future work. We add a discussion on possible future directions in Appendix H.
>
> Q3: Some typos appear as I walk through the manuscript.
>
> A3: We carefully do thorough proofreading and submit the revised version.

---

> > ### Comment · Reviewer_SmbS · 2022-11-21
> > **Thank you for your feedback**
> >
> > I appreciate your thoughtful feedback. Meanwhile, I have also read the authors' responses to other review comments.
> >
> > The authors addressed most of my concerns, especially the additional analysis of the scalability and sensitivity of hyperparameters are helpful for better understanding the proposed graph refinement method. However, the point towards the limited amount of graphs with sensitive attributes in practice raised during the discussion between the authors and K1rB seems an important concern regarding motivation.
> >
> > Thus, I would like to keep my score.

---

> > > ### Author Response · Authors · 2022-11-21
> > > **Response to Reviewer SmbS [Misunderstandings on the limited graph dataset and our motivations]**
> > >
> > > Thanks to the reviewer for the constructive comments to improve the quality of this work. We try our best the tackle reviewers' concerns. As for the concern from reviewer K1rB on "limited amount of graphs with sensitive attributes", we would like to clarify that there are **misunderstandings** for this concern, especially regarding the graph dataset and the motivation of our work.
> > >
> > > * **Graph datasets with sensitive attributes**: we agree that the number of **public** graph datasets with sensitive attributes (available sensitive attributes for all nodes) is limited in current fair graph learning research. The main reason is that the sensitive attribute of each node (person in the social network) is hard to collect due to privacy constraint. Additionally, we **mainly focus** on the scenario in that the sensitive attribute of **all** nodes is available, which makes such datasets harder to collect. The more practical situation is that only the sensitive attribute of a part of nodes is available. We would like to clarify that our main contribution is to theoretically investigate why GNN achieves higher bias than MLP as a pilot study, and we leave the **extension to limited available sensitive attributes for future work**. Meanwhile, there are many other public tabular data with sensitive attributes, such as Adult, German, and Credit. However, there is no topology information in these datasets. In this work, we follow previous work [A, B, C] to conduct experiments in fair graph learning. To the best of our knowledge, these datasets are the only available public graph datasets with sensitive attributes. The limited public graph datasets exist for current fair graph learning research, which indicates that more public graph data with sensitive attributes would be valuable. However, there is **no relationship between this concern and the motivation of our work**.
> > > * **Motivation**: The phenomenon that GNNs achieve higher bias than MLP is well-known and empirically observed in current fair graph learning literature. However, most of previous works mainly focus on how to mitigate bias for graph learning. The **reason behind such a phenomenon** is still unclear. Motivated by such observation, as a pilot study, we theoretically investigate why such a phenomenon happens. Specifically, under contextualized stochastic block model, we characterize a sufficient condition on statistical information of graph data to guarantee the phenomenon happens. We believe that the motivation of this work is clear, and hope reviewers focus on our main contribution parts.
> > >
> > > In summary, we agree that there are limited graph datasets with sensitive attributes due to privacy constrain. Many graph datasets, such as Cora, Reddit, and OGB, can not be applied in fair graph learning research since there are no sensitive attributes of each node. However, such an issue is mainly related to the **entire** fair graph learning research and indicates that more graph datasets with sensitive attributes are valuable. In our work, we don't claim the contribution for providing dataset resources, and this part is **not related to the motivation** of our work. The main contribution is to, as a pilot study, theoretically investigate why such a phenomenon happens. We hope reviewers can focus on our main contribution parts.
> > >
> > >
> > > [A] Enyan Dai and Suhang Wang. Say no to the discrimination: Learning fair graph neural networks with limited sensitive attribute information. In Proceedings of the 14th ACM International Conference on Web Search and Data Mining, pp. 680–688, 2021.
> > >
> > > [B]Ö. D. Köse and Y. Shen*, "Fairness-aware Graph Contrastive Learning," IEEE Transactions on Signal and Information Processing over Networks, May 2022.
> > >
> > > [C]Zhimeng Jiang, Xiaotian Han, Chao Fan, Fan Yang, Ali Mostafavi, and Xia Hu. Generalized demographic parity for group fairness. In International Conference on Learning Representations, 2022.

---

### Author Response · Authors · 2022-12-02
**Rebuttal Summary**

We thank the efforts of all reviewers and AC in the paper review. During the rebuttal, we try our best to address their concerns regarding the clarifications and experiments. Based on the current discussion, we believe that most of the critical concerns are addressed. The remaining concerns (from reviewer K1rB) are mainly about the strong performance of MLP and the motivation, and we provide our response one by one during the rebuttal. We are **looking forward to discussing the remaining concerns** and getting feedback from the reviewers and AC.

We provide a brief summary of the rebuttal as follows:

**[C1] Reviewer SmbS wonders about the scalability and sensitivity of hyperparameters of FairGR. Reviewer SmbS also requests more details of experimental setups, especially for the synthetic experiments.**

We add computation analysis (in Section 5) and the sensitivity of hyperparameters (in Appendix~G.2). Additionally, we also add more details on the experimental setup to make it more clear. Reviewer SmbS satisfy our revision and agree with the concern from Reviewer K1rB on our motivation. However, we could like to clarify that there is a misunderstanding and provide a detailed response in the rebuttal. Looking forward to more discussion.

**[C2] Reviewer qbDf requests more justifications on the random graph generation model, more discussion about the optimization properties of (2), and a comparison with persistency homology.**

We provide more justification for the adopted random graph generation model and comprehensive comparison with persistency homology. Our paper mainly focuses on, as a pilot study, theoretical analysis between fairness/bias and graph topology. For other graph generation models (including topology and feature distributions) and optimization properties of (2), we admit that these suggestions are good follow-up work and are left for future work. We believe that our responses tackle these concerns although there is no feedback from reviewer qbDf.

**[C3] Reviewer K1rB requests more experiments in Table. 1, and have a concern about the novelty of our paper (especially the comparison with GNN's concentration property).**

We conduct more experiments and report the performance of the proposed method "*-GR". We also provide a comprehensive comparison between GNN's concentration and bias enhancement phenomenon in GNNs., including the differences (definition and aggregation) and connections. Reviewer K1rB agrees that these concerns are tackled.

**[C4] Reviewer K1rB still has a concern about the motivation since MLP achieves good tradeoff performance, and does not agree that we use these three datasets "Pokec-n", "Pokec-z", and "NBA" since  "limited graph datasets with sensitive attributes".**

We provide detailed responses for these two concerns and we believe that there is a misunderstanding regarding the motivation, and the adopted graph datasets are standard. We are looking forward to more discussion with reviewers K1rB and SmbS.

Firstly, MLP can achieve good tradeoff performance due to the well-known (empirically observed) phenomenon that GNNs achieve higher bias than MLP. Our work is to theoretically investigate the reason behind such a phenomenon from a topology perspective. We believe that the observation (MLP achieves good tradeoff performance) supports the motivation of our work.

As for the adopted three datasets, we follow several previous works to conduct the experiments. Many graph datasets, such as Cora, Reddit, and OGB, can not be applied in fair graph learning research since there are no sensitive attributes of each node. We agree that there are limited graph datasets with sensitive attributes due to privacy constrain. However, to the best of our knowledge, the adopted datasets are the only available public graph dataset with sensitive attributes, and more public graph datasets with sensitive attributes are valuable. The limited graph datasets issue is mainly related to the entire fair graph learning research and this part is not related to the motivation of our work.

**[C5] Reviewer bTV8 has concerns about the incomplete content, the equation (2), and more experiments in Table. 1.**

We add more contents in Section 4, reorganize equation (2), and add more experimental results in Table. 1. Reviewer bTV8 admits that there is some misunderstanding about our paper, and re-evaluates our work. We believe that the concerns are addressed and thank reviewer bTV8 for voting positively on our work.

---

### Decision · Program_Chairs · 2023-01-20

**Decision:**

Reject

**Justification For Why Not Higher Score:**

Overall, the paper lacks significantly in clarity, as denoted by the reviewers in area chair, and as denoted by the new content introduced in the responses to comments (where we learn for instance that the sensitive features are not observed or used for classification). The analysis provided by the authors does not match the setting that they intend to study. Perhaps, as a side note, the problem is that the problem is not mathematically justified: the authors do not state what the "bias" is, nor do they provide  a mathematical characterization of the problem in terms of confounding variables,  unobserved variables or else --- this is perhaps what induces confusion in the reviewer strongly opposed to the rejection (and to some extent, myself) and led the authors to choose what seems to be an inappropriate theoretical analysis.

**Justification For Why Not Lower Score:**

N/A

**Metareview: Summary, Strengths And Weaknesses:**

__Summary.__ This paper considers the problem of bias amplification with Graph Neural Networks. More specifically, the authors consider the case of binary “sensitive features”. It is a known fact that in many homophilic networks, similar nodes tend to connect to one another (homophily)--- and in particular, nodes with similar sensitive features are more likely to be connected (a fact that the authors call topology bias). This, the authors argue, is detrimental to the fairness of GNNs, as it yields “representational bias" --- we note here that this phenomenon is not mathematically characterized in the paper.

From a theoretical perspective, the authors propose the analysis of a stochastic block model with two communities to show the effect of this special type of homophily. This allows them to establish conditions regarding the number of nodes and the properties of the stochastic block mode and the “reduction coefficient” of the distance between mean nodes. The authors then proceed to propose a pre-processing step to solve this issue. The pre-processing step is basically a rewiring of the adjacency matrix, obtained by solving an optimization problem that roughly, finds the closest approximation of the original adjacency $A$ that diminishes homophily on the sensitive attributes while preserving the homophily on the labels. The authors subsequently perform one synthetic experiment, that confirms their theorems, and experiments on 3 real datasets to show improved fairness metrics that their method brings.

__Summary of the reviews.__ The reviewers (and the Area Chair) seem mildly enthusiastic about this submission --- more specifically, three reviewers vote for weak acceptance (one of these reviewers being quite foreign from the field of GNNs, as they state). One reviewers votes for strong rejection (3). Although the authors dispute this reviewer's comments, I have to concur that the reviewer's feedback is pertinent.  The arguments brought forward by the reviewers were as follows:

 __1. Scalability of the method and sensitivity of the hyperparameters:__
 The reviewers questioned in particular the scalability and sensitivity of hyperparameters of FairGR – a question that was adequately addressed by the authors during the rebuttal. The method, as per the authors response, scales quadratically with the number of training samples. This, the authors argue, is not optimal --- but theirs is a “pilot” study, and therefore should be taken as a proof of concept (the method should thus not entirely be judged on its inefficiency) .

 __2. Lack of experiments:__ Several reviewers expressed the wish to see further evaluation of the method on real datasets. The authors argue that this is difficult, due to the lack of access to “sensitive” data in real datasets. This might be a fundamental limit to the authors’ ability to assess the method. However, I would like to highlight that this could warrant using more synthetic experiments to prove their claim.

 __3. Lack of novelty.__ The reviewer voting against the acceptance of the paper raises the issue that the work is not entirely novel. Indeed, the theoretical analysis of the authors is done in the case where the features are all sensitive features. In that case, the analysis is the same as what has been performed in previous work. While the authors responded to this claim in the negative, their response is unsatisfactory for  the following reasons. The authors contend that the sensitive features are usual not taken into account in GNNs  (they are unobserved --- which was not clear from the main text). I assume that this means that the sensitive features explains the connectivity patterns more than the "normal" features alone. This is not consistent with their  theoretical analysis, in which sensitive features are linked with node and label attributes --- which, here, as the reviewer highlights, is a fairly standard setting. The analysis is therefore not novel,

 __4. Unclear notations and writing (somehow alleviated after the rebuttal) noted by at least two reviewers.__ The authors seem to have considerably improved the main text though after the rebuttal. Yet, some ambiguities persist. The description of the Gaussian mixture model remains unclear: What does $\mathbb{E}_i[\mathbb{P}[s_i =1]$ mean? On what random variable is the expectation? The captions should also include the definition of the metrics used.